# ESCO1 and CTCF enable formation of long chromatin loops by protecting cohesin[STAG1] from WAPL

Gordana Wutz[1], Rene Ladurner[1†], Brian Glenn St Hilaire[2,3,4], Roman R Stocsits[1], Kota Nagasaka[1], Benoit Pignard[1], Adrian Sanborn[2,5], Wen Tang[1], Csilla Várnai[6,7], Miroslav P Ivanov[1‡], Stefan Schoenfelder[6§], Petra van der Lelij[1], Xingfan Huang[2,8,9], Gerhard Dürnberger[10#], Elisabeth Roitinger[10], Karl Mechtler[1,10], Iain Finley Davidson[1], Peter Fraser[6,11], Erez Lieberman-Aiden[2,3,4,8,12,13]*, Jan-Michael Peters[1]*

[1]Research Institute of Molecular Pathology (IMP), Vienna Biocenter (VBC), Vienna, Austria; [2]The Center for Genome Architecture, Baylor College of Medicine, Houston, United States; [3]Department of Molecular and Human Genetics, Baylor College of Medicine, Houston, United States; [4]Center for Theoretical Biological Physics, Rice University, Houston, United States; [5]Department of Computer Science, Stanford University, Stanford, United States; [6]Nuclear Dynamics Programme, The Babraham Institute, Babraham Research Campus, Cambridge, United Kingdom; [7]Centre for Computational Biology, University of Birmingham, Birmingham, United Kingdom; [8]Departments of Computer Science and Computational and Applied Mathematics, Rice University, Houston, United States; [9]Departments of Computer Science and Genome Sciences, University of Washington, Seattle, United States; [10]Institute of Molecular Biotechnology, Vienna Biocenter (VBC), Vienna, Austria; [11]Department of Biological Science, Florida State University, Tallahassee, United States; [12]Broad Institute of MIT and Harvard, Cambridge, United States; [13]Shanghai Institute for Advanced Immunochemical Studies, Shanghai Tech University, Shanghai, China

*For correspondence: erez@erez.com (EL-A); Jan-Michael.Peters@imp.ac.at (J-MP)

Present address: [†]Department of Biochemistry, Stanford University School of Medicine, Stanford, United States; [‡]The Francis Crick Institute, London, United Kingdom; [§]Epigenetics Programme, The Babraham Institute, Babraham Research Campus, Cambridge, United Kingdom; [#]Gregor Mendel Institute of Molecular Plant Biology Austrian Academy of Sciences, Vienna, Austria

**Competing interests:** The authors declare that no competing interests exist.

**Abstract** Eukaryotic genomes are folded into loops. It is thought that these are formed by cohesin complexes *via* extrusion, either until loop expansion is arrested by CTCF or until cohesin is removed from DNA by WAPL. Although WAPL limits cohesin's chromatin residence time to minutes, it has been reported that some loops exist for hours. How these loops can persist is unknown. We show that during G1-phase, mammalian cells contain acetylated cohesin[STAG1] which binds chromatin for hours, whereas cohesin[STAG2] binds chromatin for minutes. Our results indicate that CTCF and the acetyltransferase ESCO1 protect a subset of cohesin[STAG1] complexes from WAPL, thereby enable formation of long and presumably long-lived loops, and that ESCO1, like CTCF, contributes to boundary formation in chromatin looping. Our data are consistent with a model of nested loop extrusion, in which acetylated cohesin[STAG1] forms stable loops between CTCF sites, demarcating the boundaries of more transient cohesin[STAG2] extrusion activity.

## Introduction

In eukaryotic interphase cells, cohesin complexes are essential for the formation and maintenance of numerous long-range chromatin *cis*-interactions (*Gassler et al., 2017*; *Hadjur et al., 2009*; *Nativio et al., 2009*; *Rao et al., 2017*; *Schwarzer et al., 2017*; *Wutz et al., 2017*). These are

thought to have both structural and regulatory functions, in the latter case by contributing to recombination and gene regulation (reviewed in *Lin et al., 2018*; *Merkenschlager and Nora, 2016*). Hi-C experiments have revealed that chromatin interactions are either enriched in genomic regions called topologically associating domains (TADs) or appear as more pronounced localized interactions which in Hi-C maps are visible as 'dots' or on the edge of TADs as 'corner peaks'. In Hi-C maps only these dots and corner peaks are referred to as loops, even though TADs are also thought to be formed by looping of chromatin (*Dixon et al., 2012*; *Nora et al., 2012*; *Rao et al., 2014*).

Most chromatin interactions that are mediated by cohesin are anchored at genomic sites that are bound by the insulator protein CTCF (*Dixon et al., 2012*; *Nora et al., 2012*; *Rao et al., 2014*) with which cohesin co-localizes genome-wide (*Parelho et al., 2008*; *Wendt et al., 2008*). Even though the sites at which long-range interactions are anchored can be hundreds of kilobases or even Megabase pairs (Mb) apart in the linear genome, the CTCF consensus motifs that are found at these sites are typically oriented towards each other, a phenomenon known as 'the CTCF convergence rule' (*de Wit et al., 2015*; *Rao et al., 2014*; *Vietri Rudan et al., 2015*). How cohesin and CTCF generate chromatin interactions in cells is unknown, but an attractive hypothesis posits that cohesin acts by extruding loops of genomic DNA until it encounters convergently oriented CTCF sites (*Fudenberg et al., 2016*; *Sanborn et al., 2015*). This hypothesis is supported by recent single-molecule experiments which have shown that in vitro cohesin forms DNA loops by extrusion in a manner that depends on NIPBL-MAU2 (*Davidson et al., 2019*; *Kim et al., 2019*). NIPBL-MAU2's main function was so far thought to be the loading of cohesin onto DNA (*Ciosk et al., 2000*) but the experiments by Davidson et al. indicate that NIPBL-MAU2 is also part of the active cohesin holo-enzyme that mediates loop extrusion (*Davidson et al., 2019*). Loops of DNA can also be extruded by condensin (*Ganji et al., 2018*), which like cohesin belongs to the family of 'structural maintenance of chromosomes' (SMC) family of complexes (*Strunnikov et al., 1993*), and which forms DNA loops in mitosis.

Cohesin is a protein complex composed of multiple subunits. Three of these, SMC1, SMC3 and SCC1 (also known as RAD21 and Mcd1) form tri-partite rings, which in replicating cells are thought to entrap newly synthesised DNA molecules to mediate sister chromatid cohesion (*Haering et al., 2008*). During quiescence (G0) and G1, cohesin is dynamically released from chromatin via the activity of the protein WAPL and has a mean chromatin residence time of 8–25 min (*Gerlich et al., 2006*; *Kueng et al., 2006*; *Tedeschi et al., 2013*). Despite cohesin's dynamic interaction with chromatin, recent experiments suggest that some chromatin loops can persist over a significantly longer timescale of several hours (*Vian et al., 2018*). How this can occur is not understood. It is possible that chromatin interactions are maintained after cohesin has been unloaded, but the observation that experimentally induced degradation of cohesin's SCC1 subunit leads to disappearance of most TADs and loops within 15 min argues against this possibility (*Wutz et al., 2017*). More plausible scenarios are that long-lived chromatin interactions are maintained by multiple short-lived cohesin complexes, or that cohesin can be protected from WAPL so that it can persist on chromatin for longer periods of time.

Precedence for the regulatability of cohesin's residence times on chromatin comes from studies of proliferating somatic cells and from meiotic cells. During S phase of somatic mammalian cells, around half of all cohesin complexes become protected from WAPL via an incompletely understood mechanism that depends on acetylation of cohesin's SMC3 subunit (*Rolef Ben-Shahar et al., 2008*; *Unal et al., 2008*) by the acetyltransferases ESCO1 and ESCO2 and on the subsequent recruitment to cohesin of the protein sororin (*Ladurner et al., 2016*; *Lafont et al., 2010*; *Nishiyama et al., 2010*). This protection from WAPL increases cohesin's chromatin residence time to many hours and enables cohesive cohesin complexes to maintain sister chromatid cohesion from S phase until the subsequent mitosis (*Gerlich et al., 2006*; *Schmitz et al., 2007*). An even more dramatic prolongation of cohesin's residence time is thought to exist in mammalian oocytes. In these cells, a meiotic form of cohesin, cohesin[REC8], establishes cohesion during pre-meiotic S phase already before birth and then maintains it, depending on the species, for months or years until meiosis is completed during oocyte maturation cycles after puberty (*Burkhardt et al., 2016*; *Tachibana-Konwalski et al., 2010*).

Long residence times of cohesin on chromatin can also be experimentally induced by depletion of WAPL. This leads to re-localisation of cohesin from a diffuse nuclear pattern into axial chromosomal domains termed vermicelli, which are thought to represent the base of chromatin interactions

(*Tedeschi et al., 2013*). This is accompanied by an increase in the number of long DNA loops and by chromatin compaction (*Gassler et al., 2017*; *Haarhuis et al., 2017*; *Wutz et al., 2017*). These findings imply that the residence time of cohesin on chromatin can be regulated in post-replicative cells in order to maintain cohesion over long periods of time, and they show that prolonging cohesin's residence time experimentally can have major effects on genome organization (but it is important to note that the stabilization of cohesive cohesin on chromatin during S and G2 phase does not detectably alter genome architecture, which implies that cohesion and chromatin looping are mediated by distinct populations of cohesin; *Holzmann et al., 2019* and references therein). However, to date, a cohesin population with long chromatin residence times has not been identified in G1 phase, where some chromatin interactions have been reported to exist for hours (*Vian et al., 2018*).

In mammalian somatic cells, SCC1 associates with the subunits STAG1 or STAG2 to form two distinct tetrameric cohesin core complexes (*Losada et al., 2000*; *Sumara et al., 2000*). Although a recent study reported differences in the chromatin localization patterns for STAG1 and STAG2 and their contributions to chromatin organization (*Kojic et al., 2018*), it is unclear if these two forms of cohesin exhibit different residence times and incompletely understood whether they play distinct roles in loop formation. Here, we show that cohesin$^{STAG1}$ and cohesin$^{STAG2}$ do indeed display functional differences, both with respect to their residence times and in their ability to structure chromatin. We find that cohesin$^{STAG1}$ complexes are more highly acetylated, interact more stably with CTCF, form larger chromatin loops, and have a longer residence time on chromatin than cohesin$^{STAG2}$ complexes, consistent with the proposed existence of long-lived chromatin loops. This stabilization of cohesin$^{STAG1}$ depends on CTCF and ESCO1. Depletion of ESCO1 also decreases the insulation between TADs, in a manner like that observed following CTCF depletion. Furthermore, we find that both proteins are important for cohesin acetylation in G1. These results indicate that ESCO1 and CTCF function together to regulate cohesin's chromatin organization activity. Our results underline that precise regulation of cohesin's residence time is key to how cells organize their genomes. They may also be of relevance for understanding the aetiology of human cancers, in which STAG2 expression is often lost (*Lawrence et al., 2014*; *Leiserson et al., 2015*; *Solomon et al., 2011*).

## Results

### ESCO1 preferentially acetylates cohesin$^{STAG1}$ during G1 phase

Acetylation of cohesin's SMC3 subunit during S phase is known to stabilize cohesive cohesin complexes on chromatin (*Ladurner et al., 2016*), but acetylated SMC3 (SMC3ac) can also be detected in quiescent (G0) cells and in cells in G1 phase, where no cohesive cohesin exists (*Alomer et al., 2017*; *Busslinger et al., 2017*; *Minamino et al., 2015*; *Rahman et al., 2015*; *Whelan et al., 2012*). To address possible functions of SMC3 acetylation in G1 phase we first analyzed which cohesin complexes are modified during the cell cycle by ESCO1 and ESCO2. In this and subsequent experiments we detected acetylated SMC3 by using monoclonal antibodies that specifically recognize SMC3 which is acetylated either singly on K106 or doubly on K105 and K106 (*Nishiyama et al., 2010*). Immunoblot analyses of chromatin fractions isolated from synchronized HeLa cells confirmed that SMC3 acetylation is detectable throughout the cell cycle, as is ESCO1, whereas ESCO2 expression is confined to S phase (*Figure 1—figure supplement 1A*; *lanes 2–5*). As predicted from these results, depletion of ESCO1 by RNA interference (RNAi) reduced SMC3 acetylation in G1, whereas depletion of ESCO2 had little effect in this cell cycle phase (*Figure 1—figure supplement 1B*; note that contrary to the situation in G1, in G2 both ESCO1 and ESCO2 contribute to SMC3 acetylation; see also *Nishiyama et al. (2010)*. As reported by *Minamino et al. (2015)*, cohesin acetylation in G1 phase is therefore predominantly mediated by ESCO1.

Unexpectedly, however, we found that cohesin$^{STAG1}$ and cohesin$^{STAG2}$ were acetylated to different extents in G1. When cohesin$^{STAG1}$ and cohesin$^{STAG2}$ were isolated from G1 chromatin fractions by immunoprecipitation using antibodies that recognize the endogenous proteins, more acetylated SMC3 was detected in the former than the latter sample (*Figure 1A*; *compare lanes 4 and 6*). These samples had been normalized to SMC3 to account for the three-fold higher abundance of cohesin$^{STAG2}$ relative to cohesin$^{STAG1}$ on chromatin (*Holzmann et al., 2019*) but similar results were also obtained when STAG1 and STAG2 immunoprecipitates from the same number of cells were

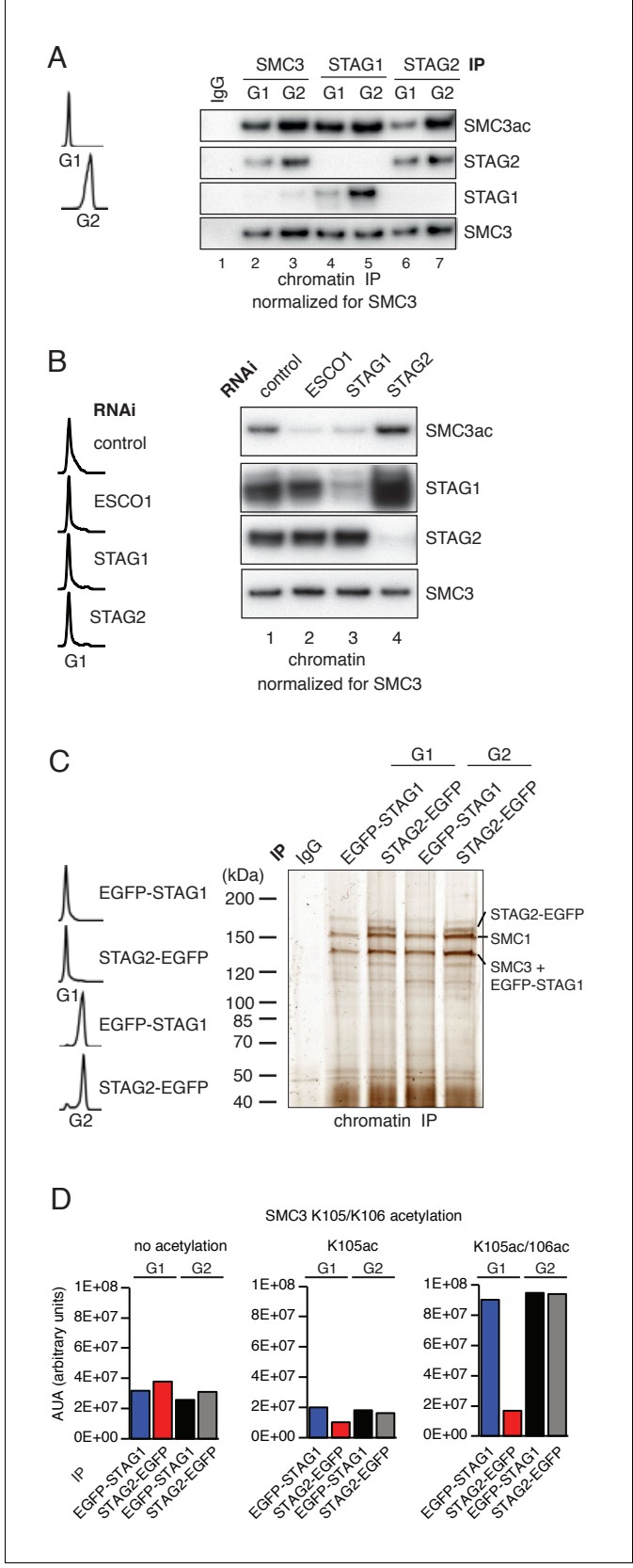

**Figure 1.** ESCO1 preferentially acetylates cohesin[STAG1] during G1 phase. (**A**) Immunoblot analysis of SMC3, STAG1 and STAG2 immunoprecipitates obtained from chromatin extracts of cells synchronized in G1 and G2. *Figure 1 continued on next page*

*Figure 1 continued*
Immunoprecipitated material was normalized to SMC3 levels and immunoblotting was performed using the indicated antibodies. Flow cytometry profiles are shown on the left. (B) Flow cytometry and chromatin extract immunoblot analysis of cells synchronized in G1 and depleted of the proteins indicated. Chromatin extracts were normalized relative to SMC3. Note that SMC3ac levels are decreased after ESCO1 and STAG1 depletion and increased after STAG2 depletion. (C) SDS-PAGE analysis of EGFP-STAG1 and STAG2-EGFP immunoprecipitates obtained from chromatin extracts of cells synchronized in G1 and G2. Immunoprecipitations were performed using anti-GFP antibodies and were analyzed with silver staining. Flow cytometry profiles are shown on the left. (D) Relative abundance of non-acetylated peptides (no acetylation), peptides acetylated at position K105 (K105ac) and peptides acetylated peptide at positions K105 and K106 (K105ac/106ac) in cells synchronized in G1 and G2 was determined by quantitative mass spectrometry from material immunoprecipitated using anti-GFP antibodies from EGFP-STAG1 and STAG2-EGFP cells. Peptide percentage was calculated relative to the total number of SMC3 peptides. Note that part of this experiment has been previously published (*Ivanov et al., 2018*; Figure S4).
The online version of this article includes the following figure supplement(s) for figure 1:

**Figure supplement 1.** ESCO1 preferentially acetylates cohesin<sup>STAG1</sup> during G1 phase.

analyzed without SMC3 normalization. Even though under these conditions more cohesin$^{STAG2}$ was present than cohesin$^{STAG1}$, less acetylated SMC3 was detected in cohesin$^{STAG2}$ than in cohesin$^{STAG1}$ (*Figure 1—figure supplement 1C*). In contrast, cohesin$^{STAG1}$ and cohesin$^{STAG2}$ contained similar amounts of acetylated SMC3 in G2 phase (*Figure 1A*; *compare lanes 5 and 7*). These results indicate that SMC3 acetylation in G1 occurs preferentially on cohesin$^{STAG1}$.

Supporting this hypothesis, we observed that STAG1 depletion by RNAi reduced SMC3 acetylation in G1 to a similar extent as depletion of ESCO1, whereas STAG2 depletion had little effect (*Figure 1B*). This experiment also revealed that more STAG1 accumulates on chromatin in STAG2-depleted cells than in control-depleted cells (*Figure 1B*, lane 4). We also observed elevated levels of STAG1 in whole cell lysates prepared from STAG2-depleted cells (*Figure 1—figure supplement 1D*, *compare lanes 1 and 10*), indicating that cells compensate for loss of STAG2 by increasing STAG1 levels by an unknown mechanism. However, this compensation is only partial as SCC1 levels were approximately three-fold lower in STAG2 depleted cells than in control cells (*Figure 1—figure supplement 1D*; compare lanes 1 and 10). This difference is important for the interpretation of Hi-C results which will be described below.

To determine the acetylation levels of cohesin$^{STAG1}$ and cohesin$^{STAG2}$, we used label-free quantitative mass spectrometry (qMS). To be able to isolate cohesin$^{STAG1}$ and cohesin$^{STAG2}$ under comparable conditions we generated HeLa cells in which all STAG1 or STAG2 alleles were modified using CRISPR-Cas9-mediated genome editing to encode enhanced green fluorescent (EGFP) fusion proteins (for characterization of these cell lines, see *Figure 2—figure supplement 1A*) and isolated cohesin$^{STAG1}$ and cohesin$^{STAG2}$ using antibodies to GFP (*Figure 1C*). This analysis indicated that in G1 cohesin$^{STAG1}$ contains four times more acetylated SMC3 than cohesin$^{STAG2}$ (*Figure 1D*).

## A subpopulation of cohesin$^{STAG1}$ associates stably with chromatin during G1 phase

Because SMC3 acetylation stabilizes cohesin on chromatin in S and G2 phase (*Ladurner et al., 2016*), we tested whether acetylated cohesin$^{STAG1}$ also has a longer residence time on chromatin in G1 than the less acetylated cohesin$^{STAG2}$ complexes. To analyze the chromatin binding dynamics of cohesin$^{STAG1}$, we performed inverse fluorescence recovery after photobleaching (iFRAP) in STAG2-depleted G1 cells that expressed a GFP-tagged version of SMC3 (SMC3-LAP; for depletion efficiency, see *Figure 2—figure supplement 1B*). Since STAG proteins are required for cohesin's association with chromatin (*Roig et al., 2014*), the behavior of SMC3-LAP in these cells should predominantly reflect the behavior of cohesin$^{STAG1}$. Conversely, to analyze cohesin$^{STAG2}$ we analyzed SMC3-LAP in cells depleted of STAG1. These experiments confirmed previous observations (*Gerlich et al., 2006*) that in control cells most cohesin interacts with chromatin dynamically with a residence of 13 min (*Figure 2A,B and E*). Similar results were obtained in cells depleted of STAG1, implying that in G1 most cohesin$^{STAG2}$ interacts with chromatin dynamically (*Figure 2A,B and E*). In contrast, in cells depleted of STAG2, the SMC3-LAP signal equilibrated much more slowly between bleached and unbleached regions (*Figure 2A and B*) and the resulting iFRAP curve could only be

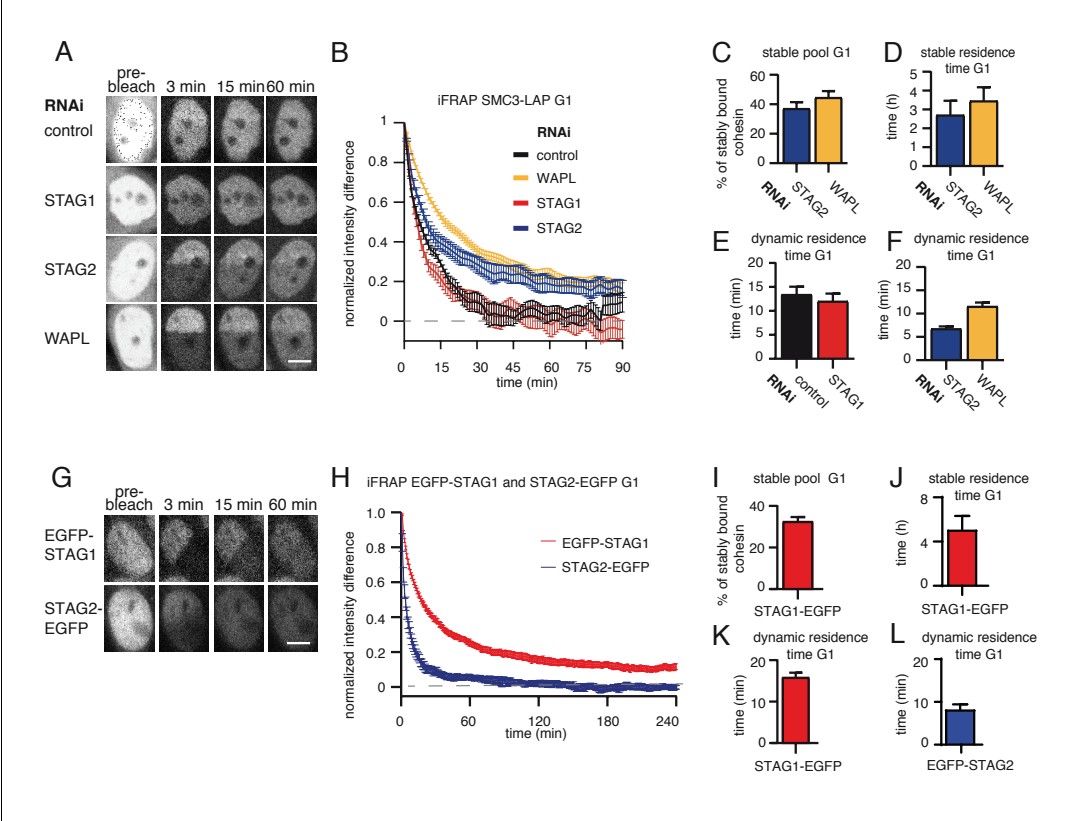

**Figure 2.** A subpopulation of cohesin[STAG1] associates stably with chromatin during G1 phase. (**A**) Images of inverse fluorescence recovery after photobleaching (iFRAP) experiments in SMC3-LAP cells synchronized in G1 and depleted of the indicated proteins by RNAi. Scale bar, 10 μm. Half of the nuclear SMC3-LAP fluorescent signal was photobleached and the mean fluorescence in the unbleached and bleached regions was monitored by time-lapse microscopy. (**B**) Graph depicting the mean normalized difference in fluorescence intensity between the bleached and unbleached regions from cells treated as described in A. Error bars denote standard error of the mean (s.e.m.), n > = 15 cells per condition. (**C**) Quantification of the fraction of nuclear SMC3-LAP that was stably chromatin bound in cells synchronized in G1 and depleted of the indicated proteins by RNAi. (**D**) Quantification of the residence time of stably chromatin bound SMC3-LAP in cells synchronized in G1 and depleted of the indicated proteins by RNAi. (**E**) Quantification of residence time of dynamically chromatin bound SMC3-LAP in cells synchronized in G1 and depleted of STAG1 by RNAi. The numbers are derived from the single exponential fit. (**F**) Quantification of the residence time of dynamically chromatin bound SMC3-LAP in cells synchronized in G1 and depleted of STAG2 and WAPL by RNAi. The numbers are derived from the bi-exponential fit. (**G**) iFRAP images in EGFP-STAG1 and STAG2-EGFP cells synchronized in G1. Scale bar, 10 μm. (**H**) Graph depicting the mean normalized difference in fluorescence intensity between the bleached and unbleached regions from cells treated as in G. Error bars denote s.e.m., n = 10 cells per condition. (**I**) Quantification of the fractions of nuclear EGFP-STAG1 that was stably chromatin bound in G1 cells. (**J**) Quantification of the residence time of stably chromatin bound EGFP-STAG1 in G1 cells. (**K**) Quantification of the residence time of dynamically chromatin bound EGFP-STAG1 in G1 cells. The numbers are derived from the biexponential fit. (**L**) Quantification of the residence time of dynamically chromatin bound STAG2-EGFP in G1 cells. The numbers are derived from the single exponential fit.

The online version of this article includes the following source data and figure supplement(s) for figure 2:

**Source data 1.** The Microsoft Excel file lists iFRAP measurements used to generate data in *Figure 2B–F*.
**Source data 2.** The Microsoft Excel file lists iFRAP measurements used to generate data in *Figure 2H–L*.
**Figure supplement 1.** Characterization of cell lines expressing tagged versions of STAG1 and STAG2 used in iFRAP curve fitting analyses.
**Figure supplement 2.** FRAP experiment confirms that subpopulation of cohesin[STAG1] associates stably with chromatin during G1 phase.
**Figure supplement 2—source data 1.** The Microsoft Excel file lists FRAP measurements used to generate data in *Figure 2—figure supplement 2*.
**Figure supplement 3.** A subpopulation of cohesin[STAG1] associates stably with chromatin in quiescent MEFs.
**Figure supplement 3—source data 1.** The Microsoft Excel file lists FRAP measurements used to generate data in *Figure 2—figure supplement 3C–F*.
**Figure supplement 3—source data 2.** The Microsoft Excel file lists FRAP measurements used to generate data in *Figure 2—figure supplement 3H–L*.

fitted to a biexponential function (*Figure 2—figure supplement 1C*). This indicates that cohesin<sup>STAG1</sup> complexes exist in two distinct populations which interact with chromatin differently. Analysis of these data revealed that most cohesin<sup>STAG1</sup> (63%) is bound to chromatin dynamically with a residence time of 7 min, but that a smaller subpopulation of cohesin<sup>STAG1</sup> (37%) is stably associated with chromatin with a residence time of 3 hr (*Figure 2C and D*). The latter residence time is similar to that observed for cohesin after depletion of WAPL (*Figure 2A–D*; *Tedeschi et al., 2013*), suggesting that the stably chromatin bound cohesin<sup>STAG1</sup> complexes are protected from release by WAPL.

To analyze the stability of cohesin<sup>STAG1</sup> and cohesin<sup>STAG2</sup> on chromatin in G1 more directly, we performed iFRAP experiments in CRISPR-generated EGFP-STAG1 and STAG2-EGFP knock-in cell lines (*Figure 2G–L*, *Figure 2—figure supplement 1A and D*). As predicted from our STAG2 depletion experiments in SMC3-LAP cells, the EGFP-STAG1 iFRAP curve could only be fitted to a biexponential function. A subpopulation (33%) of EGFP-STAG1 was bound to chromatin stably (*Figure 2I*) with a residence time of 5 hr (*Figure 2J*). The remaining 67% of EGFP-STAG1 bound dynamically to chromatin with a residence time of 15 min (*Figure 2K*). In contrast, the STAG2-EGFP iFRAP curve could be fitted to a single exponential function, with a dynamic residence time of five minutes (*Figure 2L*). A stably bound fraction of EGFP-STAG1 but not of STAG2-EGFP could also be detected in FRAP experiments, in which, compared to iFRAP, much smaller nuclear volumes are photobleached and therefore fluorescence recovery occurs on shorter time scales, resulting in higher temporal resolution (*Figure 2—figure supplement 2*). Importantly, these differences between STAG1 and STAG2 were not caused by N-terminal tagging of STAG1 since we obtained similar results in an independent cell line, in which STAG1 was C-terminally tagged with EGFP (*Figure 2—figure supplement 2*; for technical reasons we were able to generate this cell line only late during this study, which is why earlier experiments were performed with a version of STAG1 which is tagged on its N-terminus, that is, different to the C-terminally tagged STAG2). These results indicate that a small subpopulation of cohesin<sup>STAG1</sup> stably associates with chromatin in G1 in HeLa cells. We suspect that previous studies (*Gerlich et al., 2006*) failed to detect these complexes because they only represent about 9% of all cohesin complexes (37% of cohesin<sup>STAG1</sup>; *Figure 2C*, which represents 25% of all cohesin; *Holzmann et al., 2019*).

Since this stable population of G1 cohesin has not previously been described, we tested whether such a population also exists in mammalian cells other than HeLa. We therefore performed Stag2 RNAi in immortalized mouse embryonic fibroblasts (iMEFs) and monitored the fluorescence recovery of Scc1-LAP (*Figure 2—figure supplement 3A–F*) or Smc1-LAP (*Figure 2—figure supplement 3G–L*) that were expressed from bacterial artificial chromosomes (BACs; *Key resource table*). In these cells, we were able to detect stably chromatin bound cohesin in G1 even without Stag2 depletion, perhaps because in iMEFs cohesin<sup>Stag1</sup> represents 33% of total cohesin (*Remeseiro et al., 2012*). We observed that approximately 20% of Scc1-LAP and 12% of Smc1-LAP were stably bound to chromatin in G1 in these cells (*Figure 2—figure supplement 3D and J*) with a residence time of 3 hr (Scc1-LAP; *Figure 2—figure supplement 3F*) and 5 hr (Smc1-LAP; *Figure 2—figure supplement 3L*). Following Stag2 depletion the stably bound fractions of cohesin increased to 35% in both cell lines, indicating that also in MEFs predominantly cohesin<sup>Stag1</sup> stably binds to chromatin (*Figure 2—figure supplement 3D and J*).

## The stable association of cohesin<sup>STAG1</sup> with chromatin depends on ESCO1 and CTCF, as does SMC3 acetylation

Since stabilization of cohesin in S and G2 depends on ESCO proteins and sororin (*Ladurner et al., 2016*; *Schmitz et al., 2007*), we tested whether the same proteins are also required for stabilization of cohesin<sup>STAG1</sup> in G1 (note that even though sororin levels are very low in G1, some sororin can be detected during this cell cycle phase; Figure S2 in *Nishiyama et al., 2010*). To this end, we depleted sororin or ESCO1 by RNAi in G1 (*Figure 3A*) and measured the recovery of EGFP-STAG1 using iFRAP. Sororin depletion did not affect the chromatin binding dynamics of cohesin<sup>STAG1</sup> (*Figure 3B and C*), as one might have expected given the low levels of sororin in G1 (*Nishiyama et al., 2010*; *Rankin et al., 2005*). This was not due to insufficient depletion of sororin because the same siRNA oligomer reduced the stable binding of EGFP-STAG1 to chromatin in G2 (*Figure 3—figure supplement 1A–E*). In contrast, depletion of ESCO1 converted most stably bound cohesin<sup>STAG1</sup> complexes into dynamic ones (*Figure 3B and C*). This result was also observed using a second siRNA oligomer

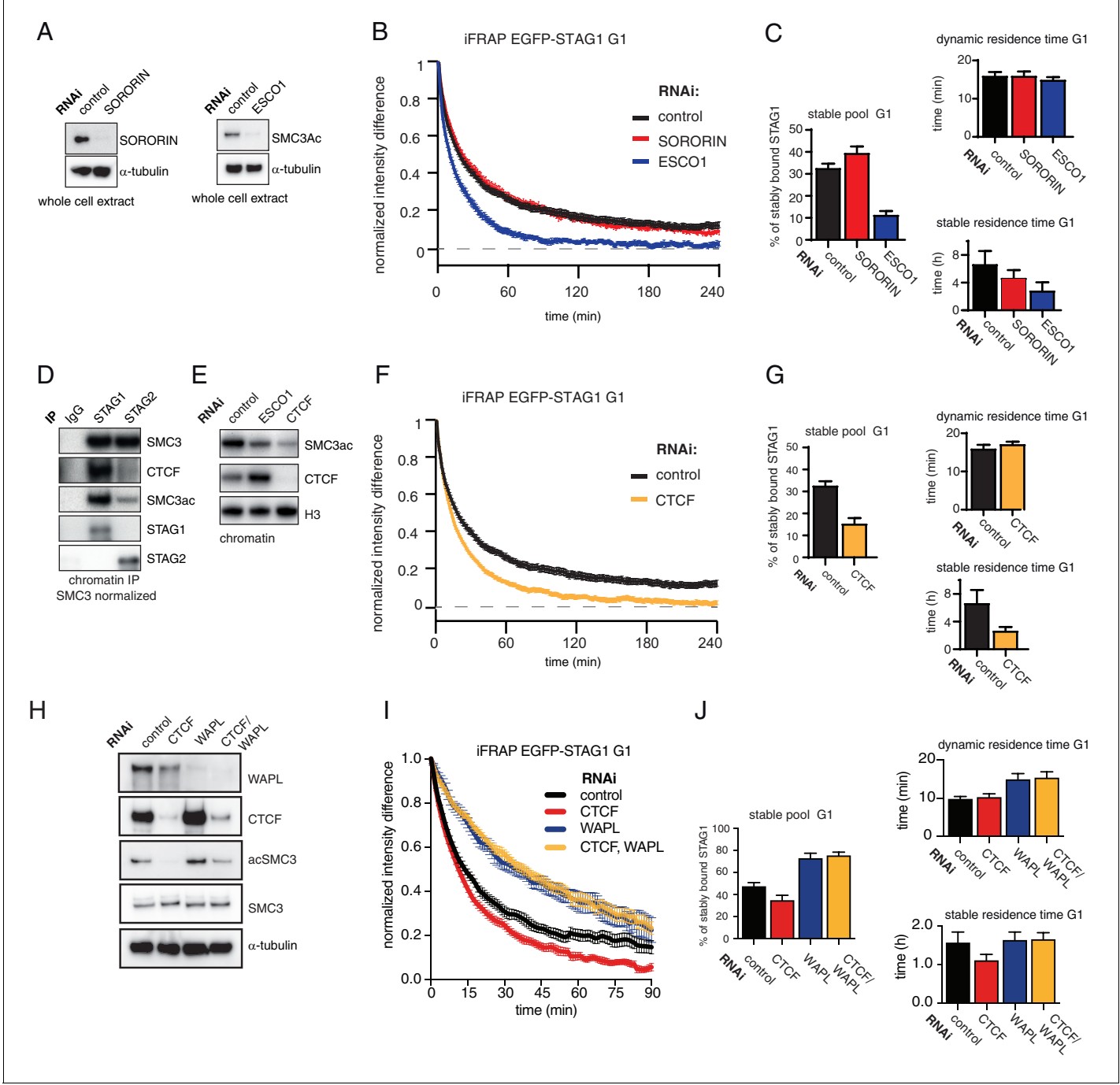

**Figure 3.** The long chromatin residence time of cohesin[STAG1] depends on ESCO1 and CTCF, as does SMC3 acetylation. (**A**) Immunoblot analysis of whole cell extract from cells depleted of sororin or ESCO1. α-tubulin was used as a loading control. (**B**) Graph depicting the mean normalized difference in EGFP-STAG1 fluorescence intensity between the unbleached and bleached regions following iFRAP in G1 cells and depletion of the indicated proteins by RNAi. Error bars denote s.e.m., n = 10 cells per condition. (**C**) Quantification of the fraction of nuclear EGFP-STAG1 that was stably chromatin bound in cells synchronized in G1 and depleted of the indicated proteins by RNAi. Quantification of dynamic and stable residence time of EGFP-STAG1 upon RNAi treatment for indicated proteins is shown on the right. (**D**) Immunoblot analysis of STAG1 and STAG2 chromatin immunoprecipitates obtained from cells synchronized in G1. Immunoprecipitations were performed using control-IgG, anti-STAG1 and anti-STAG2 antibodies. Immunoprecipitated material was normalized to SMC3 levels and immunoblotting was performed using the indicated antibodies. (**E**) Immunoblot analysis of chromatin extracts from control-, ESCO1- and CTCF-depleted HeLa cells synchronized in G1. Immunoblotting was performed using the indicated antibodies. Note that acetylation levels were decreased following depletion of CTCF. Histone H3 was used as a loading control. (**F**) Graph depicting the mean normalized difference in EGFP-STAG1 fluorescence intensity between the bleached and unbleached regions following iFRAP in cells synchronized in G1 and depleted of the indicated proteins by RNAi. Error bars denote s.e.m., n = 10 cells per condition. (**G**) Quantification of

*Figure 3 continued on next page*

*Figure 3 continued*

the fraction of nuclear EGFP-STAG1 that was stably chromatin bound in cells synchronized in G1 and depleted of the indicated proteins by RNAi. Quantification of dynamic and stable residence time of EGFP-STAG1 is shown on the right. (H) Immunoblot analysis of whole cell extract from cells depleted of CTCF, WAPL or CTCF/WAPL double depletion. α-tubulin was used as a loading control. (I) Graph depicting the mean normalized difference in EGFP-STAG1 fluorescence intensity between the unbleached and bleached regions following iFRAP in G1 cells and depleted of the indicated proteins by RNAi. Error bars denote s.e.m., n = 10 cells per condition. (J) Quantification of the fraction of nuclear EGFP-STAG1 that was stably chromatin bound in cells synchronized in G1 and depleted of the indicated proteins by RNAi. Quantification of dynamic and stable residence time of EGFP-STAG1 upon RNAi treatment for indicated proteins is shown on the right. Note that analyses of the data in I yielded lower values for stable residence times than those in C and G, presumably as a result of the shorter iFRAP imaging time used in this experiment.

The online version of this article includes the following source data and figure supplement(s) for figure 3:

**Source data 1.** The Microsoft Excel file lists iFRAP measurements used to generate data in *Figure 3B,C,F,G*.

**Source data 2.** The Microsoft Excel file lists iFRAP measurements used to generate data in *Figure 3I–J*.

**Figure supplement 1.** The long chromatin residence time of cohesin[STAG1] in G1 depends on ESCO1.

**Figure supplement 1—source data 1.** The Microsoft Excel file lists iFRAP measurements used to generate data in *Figure 3—figure supplement 1B-E*.

**Figure supplement 1—source data 2.** The Microsoft Excel file lists iFRAP measurements used to generate data in *Figure 3—figure supplement 1G-J*.

**Figure supplement 2.** CTCF is enriched in cohesin[STAG1] immunoprecipitation Volcano plots of label-free qMS data, representing protein abundance in STAG2 immunoprecipitates relative to protein abundance in STAG1 immunoprecipitates.

**Figure supplement 3.** CTCF prolongs the residence time of cohesin on chromatin.

**Figure supplement 3—source data 1.** The Microsoft Excel file lists FRAP measurements used to generate data in *Figure 3—figure supplement 3*.

targeting ESCO1 (*Figure 3—figure supplement 1F–J*), suggesting that acetylation of cohesin[STAG1] complexes is required for their stable association with chromatin.

Unexpectedly, while characterizing cohesin[STAG1] and cohesin[STAG2] immunoprecipitates, we noticed that CTCF was more abundant in cohesin[STAG1] samples, as detected by immunoblotting (*Figure 3D*) and label-free quantitative mass spectrometry (*Figure 3—figure supplement 2*). We therefore tested whether CTCF is also required for stable binding of cohesin[STAG1] to chromatin. To our surprise, CTCF depletion indeed reduced stable chromatin binding of EGFP-STAG1 to a degree like that observed following ESCO1 depletion (*Figure 3F and G*). Co-depletion of WAPL with CTCF reverted this effect (*Figure 3H–J*), indicating that CTCF contributes to long chromatin residence times of cohesin[STAG1] by protecting it from WAPL.

Similar results were obtained by performing FRAP in a cell line in which STAG1 was tagged with EGFP and STAG2 was tagged with red fluorescence protein (RFP) at their endogenous loci (*Figure 3—figure supplement 3*; for a characterization of this cell line see *Figure 2—figure supplement 1A*). Also, in this cell line, STAG1 stability on chromatin was significantly reduced following CTCF depletion (*Figure 3—figure supplement 3A and B*). The dynamic residence time of STAG2 also decreased from 5 min to 3 min (*Figure 3—figure supplement 3A and C*), indicating that CTCF also prolongs the residence time of cohesin[STAG2], although to a lesser extent than the residence time of cohesin[STAG1].

Interestingly, we found that depletion of CTCF also strongly reduced SMC3 acetylation levels (*Figure 3E*), as we had previously observed in primary Ctcf 'knockout' MEFs arrested in G0; see Figure 1A in *Busslinger et al. (2017)*. This indicates that CTCF is also required for cohesin acetylation in G0 and G1.

## Acetylated cohesin is enriched at loop anchors

Cohesin's residence time on chromatin is thought to determine the lifetime and distance of chromatin interactions (*Fudenberg et al., 2016*; *Gassler et al., 2017*; *Haarhuis et al., 2017*; *Wutz et al., 2017*). Because some cohesin[STAG1] complexes have long chromatin residence times, we analyzed whether cohesin[STAG1] complexes contribute to chromatin architecture differently during G1 than cohesin[STAG2] complexes, which have short residence times.

For this purpose, we first analyzed in ChIP-seq experiments where in the genome cohesin[STAG1] is enriched compared to cohesin[STAG2]. To avoid artefacts caused by using different antibodies, we used GFP antibodies in cell lines in which endogenous STAG1 and STAG2 had been tagged with EGFP (*Figure 2—figure supplement 1A*). This revealed that most STAG1 and STAG2 peaks

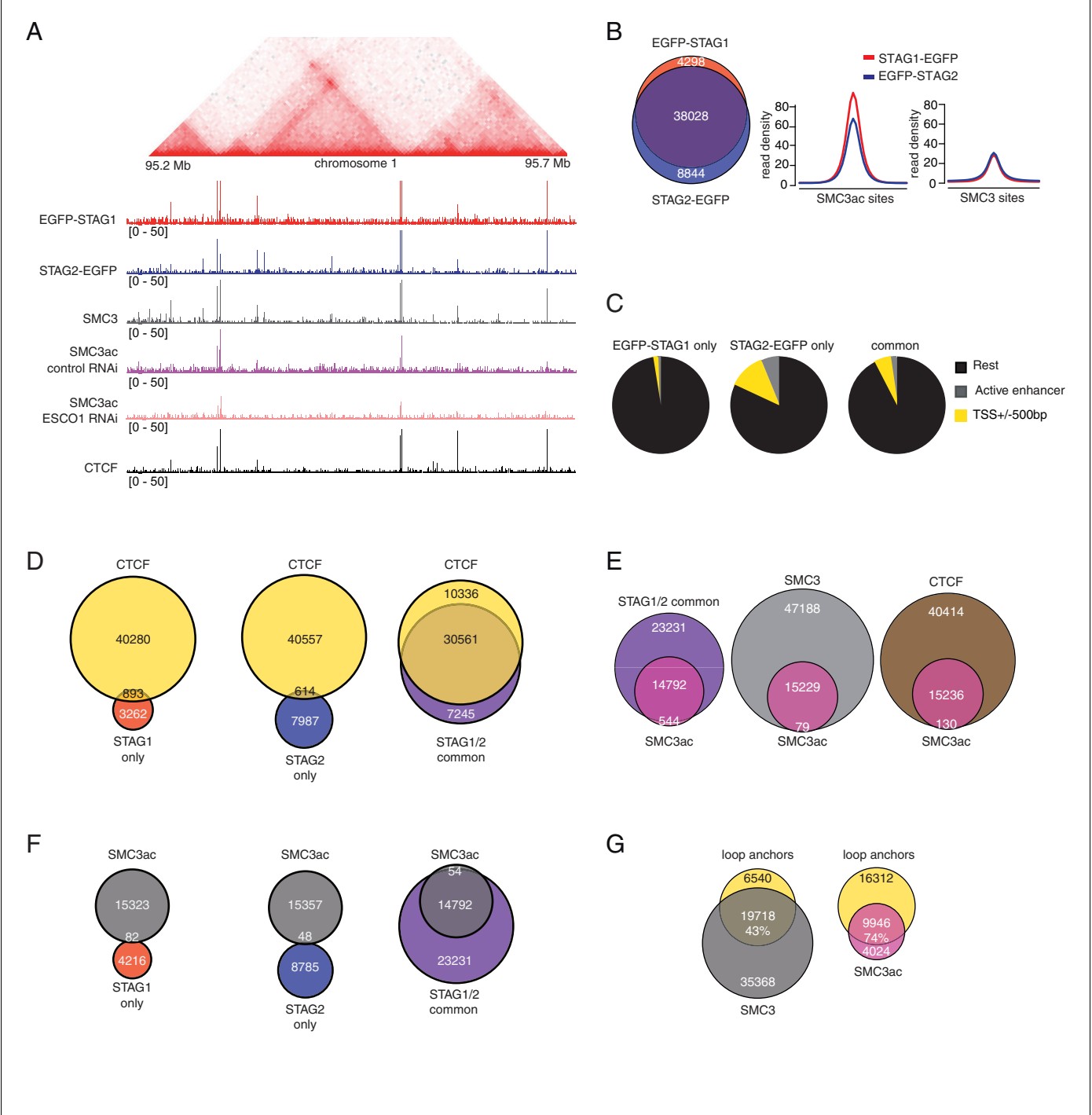

**Figure 4.** ChIP-seq analysis of the genomic distribution of STAG1, STAG2 and SMC3ac. (**A**) Coverage-corrected Hi-C contact matrix of the 95.2–95.7 Mb region of HeLa chromosome 1. ChIP-seq signals for EGFP-STAG1, STAG2-EGFP, SMC3, SMC3ac and CTCF are shown below the Hi-C contact matrix. STAG1 and STAG2 immunoprecipitation was performed using anti-GFP antibodies. Note that the SMC3ac signal is decreased following ESCO1 depletion, confirming the specificity of the anti-SMC3ac antibody. (**B**) Venn diagram illustrating the genome-wide co-localization between STAG1 and STAG2 ChIP-seq signals. STAG1/2 immunoprecipitations from EGFP-STAG1 and STAG2-EGFP cells were performed using anti-GFP antibodies. Average read density plots for EGFP-STAG1 and STAG2-EGFP at SMC3ac and SMC3 sites is shown on the right. (**C**) Pie charts showing the distribution of STAG1-only (left), STAG2-only (middle) and STAG1/2 common ChIP-seq sites (right) in G1 relative to TSS and active enhancers. (**D**) Venn diagrams showing the distribution of STAG1-only (left), STAG2-only (middle) and STAG1/2 common ChIP-seq sites (right) relative to CTCF ChIP-seq sites. (**E**) Venn diagrams illustrating the genome-wide overlap between SMC3ac and STAG1/2 common sites, SMC3 and CTCF ChIP-seq sites respectively. (**F**) Venn diagrams showing the distribution of STAG1-only (left), STAG2-only (middle) and STAG1/2 common ChIP-seq sites (right) relative to SMC3ac

*Figure 4 continued on next page*

Figure 4 continued

ChIP-seq sites. (G) Venn diagrams illustrating the genome-wide overlap between the loop anchors determined using the Hi-C map shown in (A) and SMC3 (left panel) and SMC3Ac (right panel) ChIP-seq binding sites.

overlapped (74% of all peaks; *Figure 4A and B*, *left panel*). As recently reported for mouse cells (*Kojic et al., 2018*), sites at which predominantly STAG2 was found overlapped more frequently with transcription start sites (TSSs; 11.9%) and enhancers (5.5%) than STAG1-only sites, of which only 2% and 0.05% overlapped with TSSs and enhancers, respectively (*Figure 4C*). *Kojic et al. (2018)* also reported that STAG1-only sites overlap with CTCF sites more frequently than STAG2-only sites. Although we observed a similar tendency, we found that most STAG1-only and STAG2-only sites did not overlap with CTCF, whereas most common sites did (*Figure 4D*).

Next, we determined where in the genome stably chromatin bound cohesin$^{STAG1}$ complexes are enriched. To address this, we took advantage of our finding that many if not all of these complexes are acetylated on SMC3 (*Figure 1*). We could detect acetyl-SMC3 at 24% of all SMC3 peaks (hereafter called pan-SMC3), at 39% of common STAG1 and STAG2 peaks and at 27% of CTCF peaks. Conversely, practically all SMC3-ac overlapped with SMC3, common STAG1 and STAG2 sites and CTCF (*Figure 4E*) but did not overlap with STAG1-only or STAG2-only sites (*Figure 4F*). At the common STAG1 and STAG2 sites at which acetyl-SMC3 could be detected, the read density of STAG1 was higher than that of STAG2 (*Figure 4B*, *right panel*), consistent with a longer residence time and thus higher enrichment of some of the cohesin$^{STAG1}$ complexes. Importantly, the number of acetyl-SMC3 peaks was reduced by ESCO1 depletion from 15,229 to 8,850, indicating that this antibody preferentially recognizes the acetylated form of SMC3 and not just unmodified SMC3 with reduced affinity (for a representative example, see *Figure 4A*).

We then used Hi-C to generate high resolution genome architecture maps in wild type HeLa cells (Hi-C map 1 in *Supplementary file 1*; 916 million unique read pairs). By comparing the ChIP-seq profiles of pan-SMC3 and acetyl-SMC3 with these maps, we observed that acetyl-SMC3 was more frequently found at loop anchors (74% of SMC3ac overlapped with loop anchors) than pan-SMC3 (43%; *Figure 4A and G*). We suspect that this difference is an underestimate of the specific enrichment of acetyl-SMC3 at loop anchors, as the pan-SMC3 profile presumably represents the sum of unmodified and acetylated SMC3, and because due to the resolution of our Hi-C maps, the loop anchors were on average 9 kb long and contained on average three SMC3 peaks and one acetyl-SMC3 peak. In other words, an even higher resolution Hi-C map would be expected to reveal an even stronger enrichment of acetyl-SMC3 at loop anchors.

## Cohesin$^{STAG1}$ generates long chromatin interactions

Our data obtained so far indicated that, in G1, ESCO1 predominantly acetylates cohesin$^{STAG1}$, that some of these complexes bind to chromatin stably in an ESCO1 dependent manner, and that acetylated cohesin is enriched at loop anchors. We therefore hypothesized that cohesin$^{STAG1}$ might be particularly important for forming long chromatin interactions.

To test this possibility, we depleted STAG1 or STAG2 by RNAi in cells synchronized in G1 (*Figure 5—figure supplement 1A*) and generated high-resolution Hi-C maps (Hi-C maps 2 and 3 in *Supplementary file 1*, >840 million unique pairs each). Following STAG1 depletion only minor changes could be observed, but following STAG2 depletion, that is in the presence of cohesin$^{STAG1}$, new loops became detectable (*Figure 5A*). This is remarkable given that our analysis of SCC1 had indicated that these cells contained three-fold less cohesin (see *Figure 1—figure supplement 1D* above). When we plotted the cumulative proportion of 7177 loops only detected in STAG2-depleted cells as a function of their length, we found that they were longer than control-specific loops of which we identified 14726 (*Figure 5B*), confirming that cohesin$^{STAG1}$ complexes form longer loops than cohesin$^{STAG2}$.

Corresponding changes could also be seen in contact probability plots (*Figure 5—figure supplement 1B*). Depletion of either STAG1 or STAG2 reduced the contact probability over short genomic distances (10–100 kb) but increased them over longer distances (>100 kb) compared to control cells, with STAG2 depletion having a much stronger effect. In control cells, the genomic distance with highest contact probability was around 200 kb, corresponding to the bulk of interactions that are

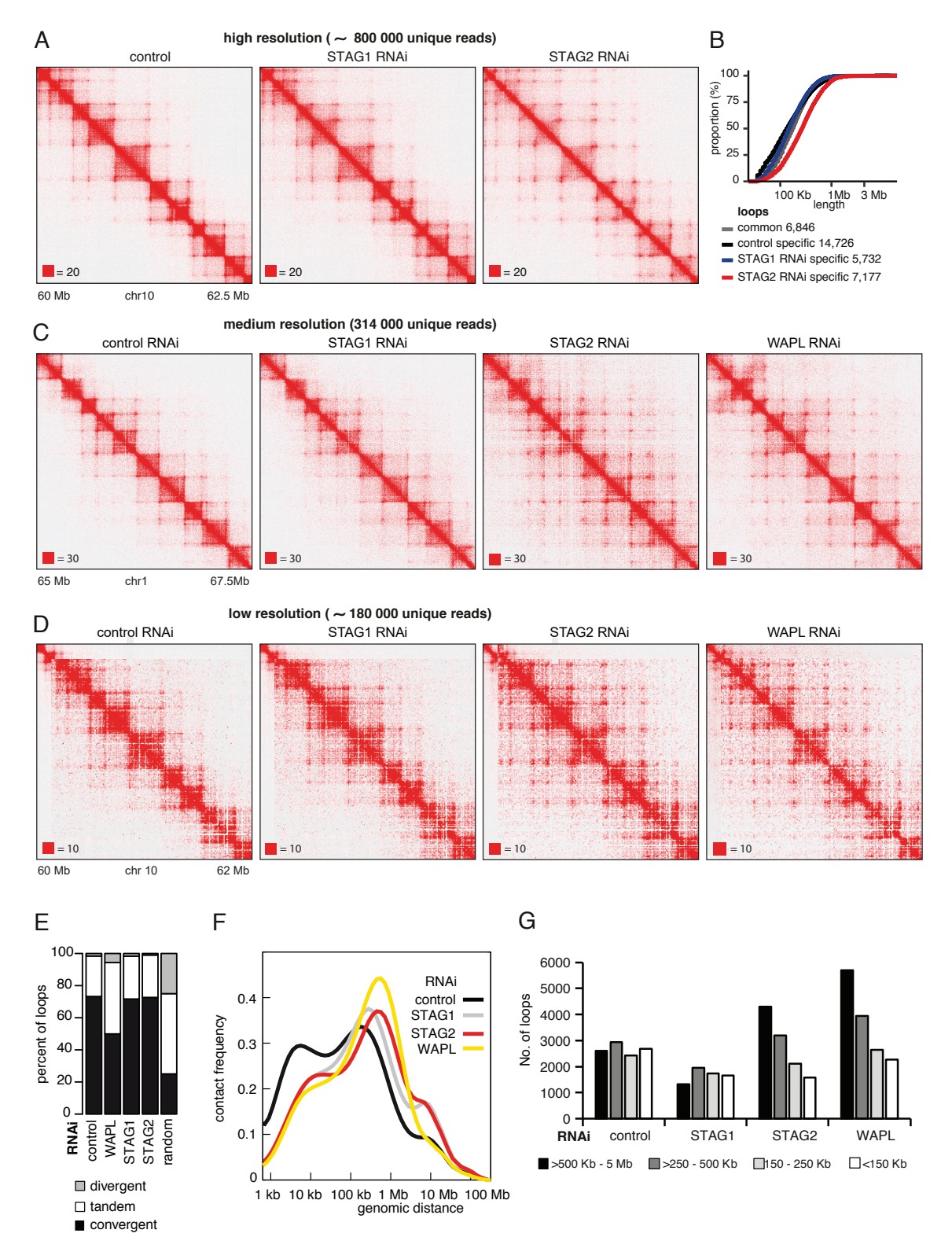

**Figure 5.** Cohesin[STAG1] generates long chromatin interactions. (**A**) Coverage-corrected Hi-C matrices of chromosome 10 (60–62.5 Mb) from control, STAG1 and STAG2-depleted cells (libraries with numbers 1,2 and 3 in *Supplementary file 1*). Matrices were plotted using Juicebox. (**B**) Cumulative distribution of loop length determined by *hiccups* in control, STAG1 and STAG2-depleted cells. (**C**) Coverage-corrected Hi-C contact matrices of chromosome 1 (65–67.5 Mb), in control-, STAG1-, STAG2- and WAPL-depleted HeLa cells (libraries with numbers 1, 2, 3 and 4 in *Supplementary file 1*

*Figure 5 continued on next page*

Figure 5 continued

were down-sampled to 314 million reads). Matrices were plotted using Juicebox. (D) Coverage-corrected Hi-C contact matrices of chromosome 10 (60–62 Mb), in control-, STAG1-, STAG2- and WAPL-depleted HeLa cells (libraries with numbers 7, 10, 11 in *Supplementary file 1*; library from the WAPL RNAi was previously published in *Wutz et al., 2017*). (E) The proportion of convergent, tandem and divergent CTCF binding orientation for loops with both anchors overlapping SMC3 and CTCF ChIP-seq peaks as well as unambiguous CTCF binding directions, for loops identified in control-depleted cells in G1-phase (ctrl, 2799 loops), in WAPL depleted but not in control-depleted cells in G1-phase (WAPL, 2295 loops), in STAG1 depleted but not in control-depleted cells in G1-phase (STAG1, 367 loops) and STAG2 depleted but not in control-depleted cells in G1-phase (STAG2, 1471 loops). The theoretically expected random proportions assuming no directionality bias are shown as comparison (7,10 and 11 in *Supplementary file 1*; library for the WAPL RNAi was previously published in *Wutz et al., 2017*). (F) Intra-chromosomal contact frequency distribution as a function of genomic distance using logarithmically increasing genomic distance bins for control-, STAG1-, STAG2- and WAPL-depleted HeLa cells. (G) Distribution of loop length in control-, STAG1-, STAG2- and WAPL-depleted HeLa cells. Loops identified by *hiccups*- (Hi-C libraries 7,10 and 11 in *Supplementary file 1*; library for the WAPL RNAi was previously published in *Wutz et al., 2017*).

The online version of this article includes the following figure supplement(s) for figure 5:

**Figure supplement 1.** Cohesin^STAG1 generates long chromatin interactions.

**Figure supplement 2.** STAG1 depletion reduces loop signal more than STAG2 depletion.

associated with TADs. This peak shifted to ~800 kb following depletion of STAG2 (*Figure 5—figure supplement 1B* and *Supplementary file 1*). In contrast, neither STAG1 nor STAG2 depletion caused major changes in TAD insulation and compartment strength (*Figure 5—figure supplement 1C and D*).

To exclude the possibility that RNAi artefacts such as off target effects could have contributed to these phenotypes, we also generated cell lines in which either STAG1 or STAG2 could be inactivated by auxin inducible degradation (AID; *Figure 5—figure supplement 1E*; note that AID tagging reduced STAG1 and STAG2 levels already in the absence of auxin, possibly due to 'leakiness' of the AID system, implying that differences between Hi-C phenotypes obtained with and without auxin treatment might be smaller than they would be otherwise). Also, in this case Hi-C experiments revealed only minor changes after STAG1 degradation, but following STAG2 depletion, that is in the presence of cohesin^STAG1, new loops became detectable (Hi-C maps 15–18 in *Supplementary file 1*, >250 million unique pairs each; *Figure 5—figure supplement 1F*). These observations indicate that cohesin^STAG1 mediates the formation of chromatin interactions that are longer than the ones formed by cohesin^STAG2.

## Fluorescence in situ hybridization experiments support the population Hi-C data obtained from STAG1-depleted and STAG2-depleted cells

To test whether our observations made by Hi-C in populations of cells could be confirmed in individual cells, we performed fluorescence in situ hybridization (FISH; *Figure 5—figure supplement 2*). We generated pairs of probes that hybridized to regions surrounding the bases of six loops with sizes of 0.8–1.5 Mb, as identified by Hi-C (*Figure 5—figure supplement 2A*, left panels), and performed FISH with each probe pair in control-, SCC1- CTCF, STAG1- STAG2- and STAG1/STAG2 - depleted cells (*Figure 5—figure supplement 2B*; see *Figure 5—figure supplement 2C* for representative images from each experimental condition). We also performed the same analysis with a probe pair not predicted to span a loop (*Figure 5—figure supplement 2A*, *last example*). We used automated image analysis to measure the three-dimensional distance between each pair of probes in more than 100 cells in each experimental condition (number of cells analyzed per condition and test of statistical significance between control and different conditions are listed in *Supplementary file 2*). This inter-probe distance was variable for each pair of probes, either reflecting technical variability, and/or indicating that the length of each predicted loop differed between cells, consistent with previous FISH and single-cell Hi-C studies (*Flyamer et al., 2017*; *Nagano et al., 2013*; *Nora et al., 2017*). In all six test loops, but not in the control genomic region, the inter-probe distance increased following depletion of SCC1 or following depletion of STAG1 in combination with STAG2. Depletion of CTCF also led to an increase in inter-probe distance in five out of six test loops, consistent with CTCF's proposed function as a boundary for loop formation. This indicates that this experimental setup can detect cohesin and CTCF-specific changes in chromatin architecture in single cells. Importantly, in five out of six test loops, depletion of STAG1 alone led to a greater increase in inter-probe distance than depletion of STAG2. This is consistent with our

hypothesis that cohesin$^{STAG1}$ is more important for generating longer-range chromatin loops than cohesin$^{STAG2}$.

## Long-range chromatin interactions mediated by cohesin$^{STAG1}$ are similar but not identical to those observed in WAPL depleted cells

Many of the loops that could only be detected in STAG2-depleted cells, that is were presumably mediated by cohesin$^{STAG1}$, had loop anchors in the outer boundaries of two or more adjacent TADs (*Figure 5A*), reminiscent of long-range chromatin interactions observed in WAPL depleted cells (*Haarhuis et al., 2017*; *Wutz et al., 2017*). We therefore compared Hi-C interactions in STAG2-depleted and WAPL-depleted cells.

For this purpose, we performed two different comparisons. In one case, we generated a new medium-resolution Hi-C map from WAPL depleted cells (Hi-C map four in *Supplementary file 1*) and compared this to the high-resolution Hi-C maps from control, STAG1 and STAG2 depleted cells (Hi-C maps 1–3 in *Supplementary file 1*; for comparability, unique sequence reads in the three latter data sets were randomly downsampled to the number of reads obtained from WAPL depleted cells; *Figure 5C* and *Figure 5—figure supplement 1E and F*). In the other case, we generated low-resolution libraries from control, STAG1-depleted and STAG2-depleted cells in replicate, using the same experimental conditions and protocols as those previously used to study the consequences of WAPL depletion (*Figure 5D*; Hi-C maps 7, 10 and 11 in *Supplementary file 1*, around 480 million read pairs each) and compared these to the previously generated replicate Hi-C maps from WAPL-depleted cells (*Wutz et al., 2017*). These comparisons revealed interesting similarities and differences between STAG2-depleted and WAPL-depleted cells.

First, we observed that most loops specifically detected in STAG2-depleted cells still 'obeyed' the CTCF convergence rule (*Figure 5E* and *Figure 5—figure supplement 1G*), whereas depletion of WAPL led to partial violation of this rule (*Figure 5E* and *Figure 5—figure supplement 1G*), as previously reported (*Haarhuis et al., 2017*; *Wutz et al., 2017*). These results indicate that cohesin$^{STAG1}$ forms long-range chromatin interactions that are anchored at convergent CTCF sites and raises the interesting possibility that WAPL is required for the CTCF convergence rule (see Discussion). Second, the genomic distance with highest contact probability was similar in STAG2-depleted cells (800 kb) and in WAPL depleted cells (900 kb), i.e., much longer than in control cells (200 kb) and in STAG1 depleted cells (300 kb; *Figure 5F*). Third, more loops > 500 kb could be detected in STAG2-depleted cells by hiccups than in control cells, similar to the situation in WAPL depleted cells where an even higher number of loops > 500 kb could be detected. In contrast, the number of loops > 500 kb was reduced in STAG1-depleted cells (*Figure 5G* and *Figure 5—figure supplement 1H*).

Together, these observations support the notion that acetylated cohesin$^{STAG1}$ complexes are protected from WAPL, therefore have longer residence times on chromatin and can form longer chromatin interactions, which, however, still obey the CTCF convergence rule.

## In silico modeling indicates that long chromatin residence time of cohesin$^{STAG1}$ causes formation of long chromatin loops

To test the hypothesis that an increased chromatin residence time of cohesin$^{STAG1}$ causes the formation of long chromatin loops we performed simulations *in silico*. For this purpose, we used a simplified hypothetical DNA sequence that contained three pairs of convergent CTCF sites and assumed three different extrusion times (*Figure 6A*). We also used molecular dynamics simulations to model the behavior of chromatin in the presence of loop extrusion complexes and generated an *insilico* contact map of a region of human chromosome 9. This map resembled the Hi-C map of this region generated from control HeLa cells (*Figure 6B and C*, compare top and bottom panels). We next simulated the effect of altering the lifetime of loop extrusion complexes on DNA and found, consistent with results by *Fudenberg et al. (2016)* , that longer lifetimes resulted in the generation of longer-range interactions, and *vice versa*. These *in silico* contact maps reproduced the changes we observed in cells following STAG1 and STAG2 depletion reasonably well. Thus, our experimental and simulation data are consistent with the hypothesis that cohesin$^{STAG1}$ and cohesin$^{STAG2}$ contribute to chromatin organization differentially, and that, by virtue of its longer residence time on chromatin, cohesin$^{STAG1}$ generates longer-range interactions than cohesin$^{STAG2}$.

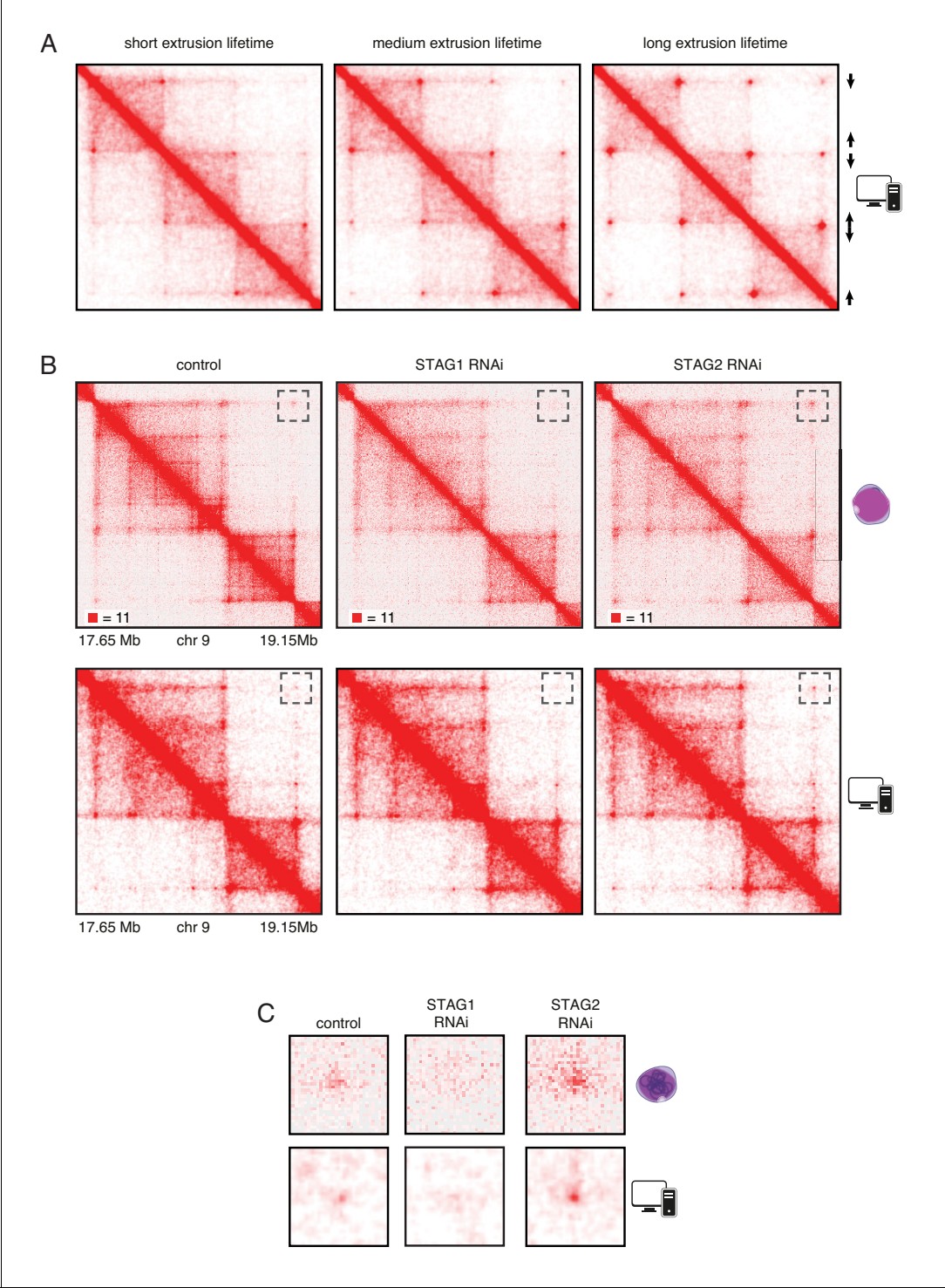

**Figure 6.** In silico modeling confirms that stably bound cohesin forms longer loops. (**A**) Simplified simulation with three pairs of convergent CTCF sites to demonstrate the relationship between extrusion lifetime and loop size. Loop strength positively correlates with increased extrusion lifetime. (**B**) Coverage-corrected Hi-C matrices of chromosome 9 (17.65–19.15 Mb) from control, STAG1 and STAG2-depleted cells (upper panels). Hi-C simulations of the same genomic region were performed with short, medium and long extrusion complex lifetimes. Longer extrusion lifetimes generated Hi-C matrices that resembled those from STAG2-depleted cells. (**C**) Zoom in of the loops that are boxed in panel B.

## ESCO1, like CTCF, regulates the loop formation activity of cohesin^STAG1

The hypothesis that stably chromatin bound cohesin^STAG1 complexes form long chromatin loops predicts that also ESCO1 and CTCF are required for these interactions, since we had found that both are required for the long residence time of cohesin^STAG1 (*Figure 3A–G*). To test this prediction, we co-depleted ESCO1 or CTCF together with STAG2, synchronized cells in G1 and performed Hi-C analysis (*Figure 7*, *Figure 7—figure supplement 1*; Hi-C maps 7, 8, 9, 11, 12 and 13 in *Supplementary file 1*, around 200 million unique read pairs each). Indeed, we found that the appearance of new loops in STAG2 depleted cells was largely reverted by co-depletion of ESCO1 or CTCF (*Figure 7—figure supplement 1C–F*). This epistatic behavior of ESCO1 and CTCF depletion over STAG2 depletion supports the hypothesis that the long residence time of cohesin^STAG1 enables the formation of long chromatin loops in G1.

We also analyzed cells from which only CTCF or ESCO1 had been depleted (*Figure 7*). Consistent with previous results obtained by AID of CTCF (*Nora et al., 2017*; *Wutz et al., 2017*), depletion of CTCF by RNAi resulted in a reduction in the number of detectable loops (*Figure 7A,C and D*), and a decrease in the TAD insulation score (*Figure 7E*) but did not abolish long-range chromatin interactions. To the contrary, contact probability analysis revealed that the interactions involved in TAD formation were longer in cells depleted of CTCF (*Figure 7F*; note that this particular effect was less pronounced after auxin induced CTCF degradation, perhaps because in these experiments CTCF levels were also reduced in control cells due to 'leakiness' of the AID system; *Wutz et al., 2017*). As concluded previously (*Nora et al., 2017*; *Wutz et al., 2017*), these results suggest that CTCF is not required for long-range chromatin interactions per se but for specifying the loop anchors which mediate these, possibly by functioning as a boundary for loop extruding cohesin complexes. Remarkably, ESCO1 depletion caused similar effects, that is a reduction in the number of detectable loops (*Figure 7A,C and D*), a decrease in the TAD insulation score (*Figure 7E*) and an increase in the length of chromatin interactions (*Figure 7F*). This raises the interesting possibility that ESCO1, like CTCF, is important for restricting cohesin's loop formation activity (see Discussion).

However, a comparison of Hi-C phenotypes between ESCO1-depleted and CTCF-depleted cells did not only reveal similarities but also differences. First, compared to both CTCF-depleted and control cells, contact frequencies were reduced in the 0–50 kb range following ESCO1 depletion (*Figure 7F*). Second, ESCO1 depletion also resulted in an increase in contact probability around 20 Mb, the genomic distance associated with compartmentalization (*Figure 7F*). Consistently, the analysis of whole chromosome Hi-C maps revealed that depletion of ESCO1, either alone or in combination with STAG2 depletion, enhanced the 'checkerboard' pattern indicative of compartmentalization and led to a strong increase in interactions around 10 Mb (*Figure 7B*). Genome-wide aggregate analysis of 50 compartment categories ranging from strong B to strong A compartments confirmed this, showing increasing contact enrichment between similar compartment categories and a decreasing contact enrichment between dissimilar (e.g., strong A and strong B) compartment bins in both long *cis* (>2 Mb) and *trans* interactions (*Figure 7—figure supplement 1G*). This phenotype is reminiscent of the increase in compartmentalization observed following cohesin depletion (*Flyamer et al., 2017*; *Rao et al., 2017*; *Schwarzer et al., 2017*; *Wutz et al., 2017*), even though ESCO1-depleted cells contained as much cohesin on chromatin as the corresponding control of STAG2-depleted cells (*Figure 7—figure supplement 1A*, compare SCC1 signals in lanes 3 and 6, and in lanes 9 and 12). These observations raise the interesting possibilities that the ability of cohesin to suppress compartmentalization depends on ESCO1, or that this acetyltransferase has additional functions in chromatin organization that are independent of cohesin.

## Discussion

### Acetylated cohesin^STAG1 complexes are protected from WAPL by CTCF and form long chromatin loops

In interphase cells, cohesin folds genomic DNA into thousands of loops which are thought to have both structural and regulatory functions. Little is known about the lifetime of these loops and how their formation and maintenance is controlled. It has generally been assumed that loops are short-lived dynamic structures because the cohesin complexes that form them interact with DNA only briefly, in mammalian cells on average for 8–25 min during G0 and G1 phase (*Gerlich et al., 2006*;

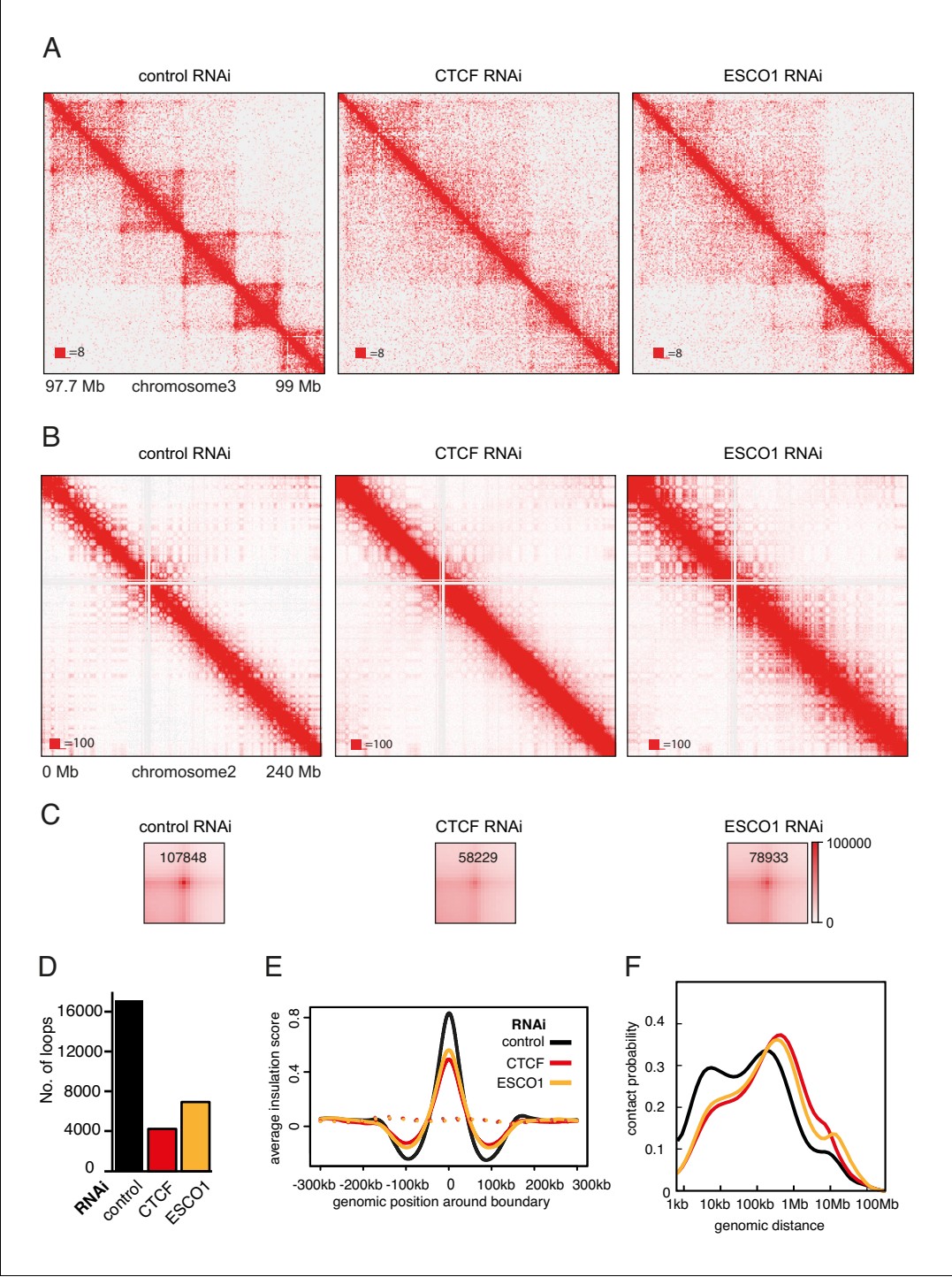

**Figure 7.** The function of long chromatin loops by cohesin[STAG1] depends on CTCF and ESCO1. (**A**) Coverage-corrected Hi-C contact matrices of chromosome 3 (97.7–99 Mb) in control-, CTCF- and ESCO1-depleted HeLa cells. Matrices were plotted using Juicebox. (**B**) Coverage-corrected Hi-C contact matrices of chromosome 2 (0–200 Mb) in control-, CTCF- and ESCO1- depleted HeLa cells. Matrices were plotted using Juicebox. (**C**) Total contact counts around loops longer than 150 kb in control-, CTCF- and ESCO1-depleted HeLa cells are shown in the lower panel. Loops were identified by *hiccups* in control cells. The values of the centre peaks are indicated. (**D**) Number of loops identified by *hiccups* in the matrices described in (**A**). (**E**) Average insulation score around TAD boundaries in control-, ESCO1- and CTCF-depleted cells synchronized in G1. Dashed lines indicate the average insulation score between random positions. (**F**) Intra-chromosomal contact frequency distribution as a function of

*Figure 7 continued on next page*

*Figure 7 continued*

genomic distance using logarithmically increasing genomic distance bins for control-, CTCF- and ESCO1-depleted HeLa cells.

The online version of this article includes the following figure supplement(s) for figure 7:

**Figure supplement 1.** Effect of ESCO1 and CTCF depletion on chromatin organization.

*Tedeschi et al., 2013*), after which they are released by WAPL (*Kueng et al., 2006*). Despite this, long-range chromatin interactions as they can be detected by Hi-C change little over time (*Nagano et al., 2017*; *Wutz et al., 2017*), and recent evidence implies that some loops can persist for hours (*Vian et al., 2018*). It is not known whether these stable structures are maintained by dynamically exchanging cohesin complexes or by an unknown mechanism that would protect cohesin from release by WAPL. Precedence for the latter scenario comes from the observation that cohesin complexes that mediate cohesion in proliferating cells are protected from WAPL by acetylation of their SMC3 subunit (*Rolef Ben-Shahar et al., 2008*; *Unal et al., 2008*), by subsequent recruitment of sororin and at mitotic centromeres also by shugoshin (*Hara et al., 2014*; *Nishiyama et al., 2010*). However, these cohesive complexes do not seem to participate in loop formation, as their stabilization on chromatin does not detectably alter chromatin structure in G2 phase (*Wutz et al., 2017*), whereas experimental stabilization of all cohesin complexes on chromatin does (*Tedeschi et al., 2013*; discussed in *Holzmann et al. (2019)*. It has therefore remained unknown whether loop forming cohesin complexes can be protected from WAPL to extend the lifetime of chromatin loops.

Here we provide evidence that such a regulatory mechanism exists in human cells, since our FRAP and Hi-C experiments have identified a small subpopulation of cohesin$^{STAG1}$ complexes that persist on chromatin for hours and contribute to the formation of long chromatin loops. Our results indicate that the stabilization of these cohesin$^{STAG1}$ complexes on chromatin depends on SMC3 acetylation, as does the stabilization of cohesive cohesin in S and G2 (*Ladurner et al., 2016*). But in contrast to cohesive cohesin, loop forming cohesin$^{STAG1}$ complexes can persist on chromatin for hours in the absence of sororin, as one might have predicted since sororin is only present in proliferating cells, and in these almost exclusively from S phase until mitosis (*Nishiyama et al., 2010*; *Rankin et al., 2005*). Likewise, we suspect that shugoshin is dispensable for the long chromatin residence time of acetylated cohesin$^{STAG1}$, since shugoshin specifically protects cohesive cohesin from WAPL at centromeres in mitosis (*Hara et al., 2014*). Unexpectedly, however, we found that CTCF is essential for the long residence time of cohesin$^{STAG1}$ on chromatin, indicating that CTCF, like sororin and shugoshin, is a WAPL antagonist that can prevent cohesin from being released from DNA (*Figure 8A*).

The notion that cohesin$^{STAG1}$ complexes are protected from WAPL by SMC3 acetylation and CTCF is supported by the observation that the chromatin residence time of these complexes is similarly long as the residence time of cohesin in WAPL depleted cells (*Figure 2*), by our finding that cohesin$^{STAG1}$ complexes form longer chromatin loops as cohesin does in the absence of WAPL (*Figure 5*), and by the epistatic effect of WAPL depletion over CTCF depletion (*Figure 3H*). Interestingly, however, the extended loops formed by cohesin$^{STAG1}$ differ in one important aspect from the loops that are formed by cohesin in the absence of WAPL, in that the former are typically anchored at convergent CTCF sites (*Figure 5E* and *Figure 5—figure supplement 1G*) whereas the latter are often anchored at tandemly oriented CTCF sites (*Haarhuis et al., 2017*; *Wutz et al., 2017*; *Figure 5E* and *Figure 5—figure supplement 1G*). This difference implies that WAPL may contribute to the CTCF convergence rule, that is may have a role in ensuring that loops are only anchored at convergent CTCF sites.

Our photobleaching experiments also revealed that most if not all cohesin$^{STAG2}$ complexes, which are three fold more abundant in HeLa cells than cohesin$^{STAG1}$ complexes (*Holzmann et al., 2019*), have short chromatin residence times in the range of minutes, implying that they are dynamically released from chromatin by WAPL. Accordingly, CTCF depletion only had a small effect on their chromatin residence time. However, this does not exclude the possibility that CTCF would also be able to protect cohesin$^{STAG2}$ from WAPL under conditions where these complexes become stabilized on chromatin.

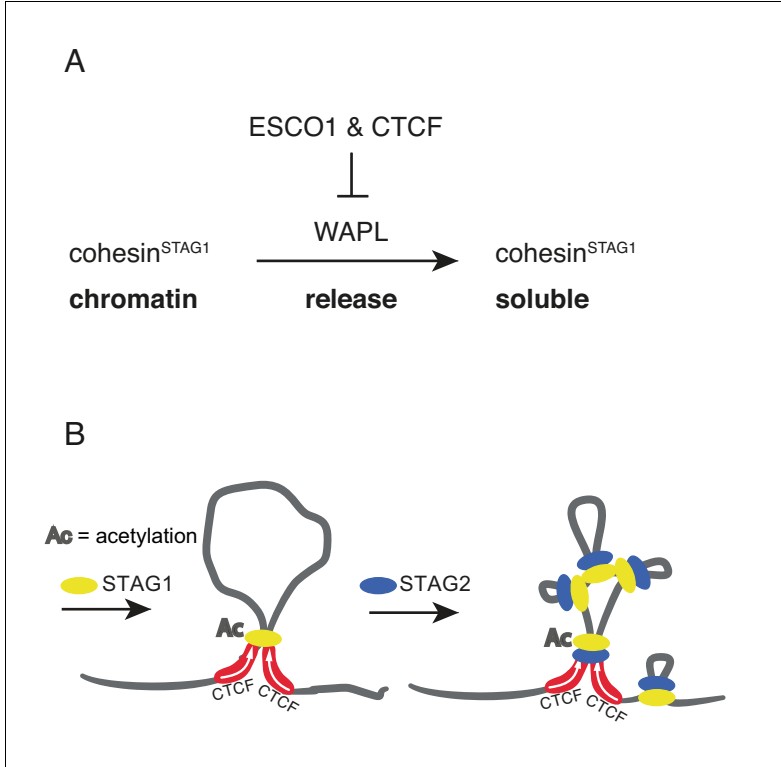

**Figure 8.** Schematic model. (**A**) ESCO1 and CTCF stabilize cohesinSTAG1 by inhibiting WAPL. (**B**) Schematic model of nested loop extrusion. CohesinSTAG1 makes longer loops; cohesinSTAG2 makes loops in and around cohesinSTAG1 loops.

The existence of two different forms of cohesin with different chromatin residence times is reminiscent of the situation in mitotic chromosomes. Their structural organization is thought to depend on the sequential action of first condensin II to form large loops and then condensin I to form smaller, nested loops (*Gibcus et al., 2018*; *Walther et al., 2018*). It is unknown how interphase chromatin architecture is established upon exit from mitosis, but recent fluorescence correlation spectroscopy experiments indicate that STAG1 is recruited to chromatin earlier than STAG2 (*Cai et al., 2018*). Given our finding that cohesinSTAG1 generates longer loops than cohesinSTAG2, it is tempting to speculate that cohesinSTAG1 and cohesinSTAG2 function analogously to condensin II and condensin I in mitosis, respectively, by forming loops within loops on interphase chromatin (*Figure 8B*).

Consistent with our findings, it has recently been reported that cohesinSTAG1 and cohesinSTAG2 contribute to chromatin differently (*Casa et al., 2019*; *Viny et al., 2019*; *Cuadrado et al., 2019*; *Kojic et al., 2018*), with cohesinSTAG1 contributing more to TAD organization than cohesinSTAG2. CohesinSTAG1 has also been shown to be more resistant to biochemical salt extraction from chromatin than cohesinSTAG2 (*Kojic et al., 2018*), but whether this property is related to the long chromatin residence time of acetylated cohesinSTAG1 reported here is unknown. Both our study and that of *Kojic et al. (2018)* provide evidence that STAG1 and STAG2 are enriched at CTCF sites and enhancers, respectively. This raises the possibility that cohesinSTAG2 might regulate promoter–enhancer interactions. STAG2 is one of only twelve genes known to be mutated in more than four major human cancer types (*Lawrence et al., 2014*). It is therefore possible that changes in gene expression following STAG2 mutation might be a common early event in human carcinogenesis.

## ESCO1 is needed to constrict cohesin at CTCF sites

The acetylation that protects cohesive cohesin from WAPL from S phase until mitosis is thought to depend on both ESCO1 and ESCO2 (*Ladurner et al., 2016*; *Nishiyama et al., 2010*). In contrast, acetylation of cohesin during G1 is only mediated by ESCO1, since ESCO2 is absent during this

phase of the cell cycle (*Alomer et al., 2017*; *Minamino et al., 2015*; *Rahman et al., 2015*; *Whelan et al., 2012*; *Figure 1—figure supplement 1A*). Our finding that ESCO1 is needed for a long chromatin residence time of a subpopulation of cohesin complexes and for their ability to form extended chromatin loops reveals for the first time a function of SMC3 acetylation in G0 and G1, and at the same time indicates that ESCO1 has a certain degree of substrate specificity, at least during G1 phase, by preferentially acetylating cohesin[STAG1].

Interestingly, our results revealed that ESCO1 might also have another function in chromatin organization. ESCO1 depletion reduced the Hi-C dots and corner peaks specifically detected after STAG2 depletion, that is long chromatin loops formed by cohesin[STAG1], as one would have predicted given that ESCO1 is required for the long residence time of a subpopulation of cohesin[STAG1]. However, in addition, ESCO1 also decreased TAD insulation, to an extent as seen after CTCF depletion. This implies that ESCO1 might have two functions in chromatin organization: protection of cohesin[STAG1] from WAPL, and a boundary function in loop extrusion. If ESCO1 has such a boundary function, this could help to explain why co-depletion of ESCO1 and ESCO2 with WAPL resulted in the formation of vermicelli in chicken DT40 cells, whereas in these cells WAPL depletion alone did not cause this phenotype (*Kawasumi et al., 2017*). It is possible that in these experiments WAPL depletion increased the residence time of cohesin on chromatin, whereas depletion of ESCO1 compromised the function of CTCF boundaries, so that extruding cohesin complexes could form longer loops and ultimately accumulate in vermicelli domains.

The hypothesis that ESCO1 contributes to boundary function could also explain why PDS5A and PDS5B are required for boundary function (*Wutz et al., 2017*) because SMC3 acetylation by ESCO1 and ESCO2 has been shown to depend on PDS5A and PDS5B (*Minamino et al., 2015*; for a similar dependency in yeast see *Chan et al., 2013* and *Vaur et al., 2012*). It is therefore possible that one function of PDS5 proteins in chromatin architecture is to facilitate SMC3 acetylation by ESCO1. In addition, PDS5 proteins might influence chromatin architecture by other mechanisms as they are also required for WAPL-mediated release of cohesin from chromatin (*Chan et al., 2012*; *Ouyang et al., 2016*; *Shintomi and Hirano, 2009*; *Wutz et al., 2017*). Interestingly, we also observed a reduction in SMC3 acetylation following depletion of CTCF, indicating that SMC3 acetylation also depends on CTCF, and that the functions of ESCO1 and CTCF at chromatin boundaries might be interdependent (*Busslinger et al., 2017*; this study).

Consistent with our results, a role for the yeast ortholog of ESCO1 and ESCO2 (Eco1) in constraining cohesin's ability to form mid-range chromatin interactions, together with a dual role for the PDS5A and PDS5B ortholog Pds5 in regulating Eco1 and Wapl was reported during preparation of this manuscript (*Dauban et al., 2020*). The molecular mechanisms through which STAG1 and STAG2 alter the properties of cohesin remain to be understood, but our results suggest that cohesin[STAG1] is more frequently stabilized at CTCF sites via PDS5A, PDS5B and ESCO1 than cohesin[STAG2]. The regulation of chromatin boundaries by cohesin acetyltransferases and PDS5 proteins may therefore be an evolutionarily conserved mechanism, which might be spatially controlled in mammalian cells by CTCF.

# Materials and methods

### Key resources table

| Reagent type (species) or resource | Designation | Source or reference | Identifiers | Additional information |
|---|---|---|---|---|
| Cell line (*H. sapiens*) | Hela Kyoto BAC recombineering SMC3-LAP | *Poser et al., 2008* ([DOI: 10.1038/nmeth.1199](https://doi.org/10.1038/nmeth.1199)) | | *Figure 2* |
| Cell line (*H. sapiens*) | Hela Kyoto CRISPR EGFP-STAG1 cl H4 | this study | | *Figure 2—figure supplement 1* |
| Cell line (*H. sapiens*) | Hela Kyoto CRISPR STAG1-EGFP H8 | *Cai et al., 2018* | | *Figure 2—figure supplement 1* |
| Cell line (*H. sapiens*) | Hela Kyoto CRISPR STAG2-EGFP F2 | *Cai et al., 2018* | | *Figure 2—figure supplement 1* |

*Continued on next page*

*Continued*

| Reagent type (species) or resource | Designation | Source or reference | Identifiers | Additional information |
|---|---|---|---|---|
| Cell line (*H. sapiens*) | Hela Kyoto CRISPR EGFP-STAG1 FLAG-RFP-STAG2 clD10 | this study | | *Figure 2—figure supplement 1* |
| Cell line (*H. sapiens*) | Hela Kyoto CRISPR STAG1-AID | this study | | *Figure 5—figure supplement 1* |
| Cell line (*H. sapiens*) | Hela Kyoto CRISPR STAG2-AID | this study | | *Figure 5—figure supplement 1* |
| Cell line (*M. musculus*) | iMEFs BAC recombineering SMC1-LAP | *Poser et al., 2008* (DOI: 10.1038/nmeth.1199) | | *Figure 2—figure supplement 3* |
| Cell line (*M. musculus*) | iMEFs BAC recombineering SCC1-LAP | *Poser et al., 2008* (DOI: 10.1038/nmeth.1199) | | *Figure 2—figure supplement 3* |
| Antibody | Anti-CTCF | Peters laboratory | Antibody ID: A992 | Western blotting |
| Antibody | Anti-CTCF | Merck Milipore | Cat# 07–729, RRID:AB_441965 | ChIP |
| Antibody | Anti-ESCO1 | Peters laboratory | Antibody ID:782M | Western blotting |
| Antibody | Anti-ESCO2 | gift from J. de Winter | | Western blotting |
| Antibody | Anti-GFP | Roche | Cat# 11814460001, RRID:AB_390913 | Western blotting |
| Antibody | Anti-GFP | Abcam | Cat# ab290, RRID:AB_303395 | ChIP |
| Antibody | phospho-histone Histone H3 (Ser10) | Cell Signaling | Cat #:9701, RRID:AB_331535 | Western blotting |
| Antibody | PCNA | Santa Cruz | Cat #:PC10, RRID:AB_628110 | Western blotting |
| Antibody | Anti-SCC1 | EMD Milipore Corporation | Cat #:05–908 RRID:AB_417383 | Western blotting |
| Antibody | Anti-SMC1 | Bethyl Laboratories | Cat #:A300-055A, RRID:AB_2192467 | Western blotting |
| Antibody | Anti-SMC3 | Peters laboratory | Antibody ID:A941 | ChIP, Western blotting |
| Antibody | Anti-Smc3 acetyl | gift from K. Shirahige | | ChIP, Western blotting |
| Antibody | Anti-sororin | Peters laboratory | Antibody ID:A953 | Western blotting |
| Antibody | Anti-STAG1 | Peters laboratory | Antibody ID:A823 | ChIP, Western blotting |
| Antibody | Anti-STAG2 | Bethyl | Cat #:A300-158A, RRID:AB_185514 | Western blotting |
| Antibody | Anti-tubulin | Sigma | Cat #:T-5168, RRID:AB_477579 | Western blotting |
| Antibody | Anti-WAPL | Peters laboratory | Antibody ID:A1017 | Western blotting |

## Cell culture, cell synchronization and RNA interference

HeLa Kyoto cells (*Landry et al., 2013*; RRID:CVCL_1922) were free from detectable mycoplasma contamination and have been authenticated by STR fingerprinting (Vienna Biocenter Core Facilities). Cells were cultured in DMEM supplemented with 10% FBS, 0.2 mM glutamine and penicillin/streptomycin (Gibco). Cells were synchronized at early S phase by two consecutive rounds of treatment with 2 mM thymidine (Sigma) and released into fresh media for 6 hr (G2) or 15 hr (G1). Synchronization was assessed by flow cytometry after methanol fixation and propidium iodide staining as described (*Ladurner et al., 2014*). Cells were treated with 30 nM siRNAs as indicated using RNAiMax

(Invitrogen) at 48 or 72 hr before downstream analyses. Pre-annealed 21 nucleotide RNA with 3' double thymidine overhangs (*Elbashir et al., 2001*) was purchased from Ambion. Sense sequences for control, CTCF ('#1') (*Wendt et al., 2008*), ESCO1, ESCO2 (*Nishiyama et al., 2010*), Wapl ('Wapl1'), STAG1 ('SA1'), STAG2 ('SA2') (*Kueng et al., 2006*), and sororin (*Schmitz et al., 2007*) were denoted previously. ESCO1 second siRNA was a pool of 4 siRNAs: GGAAAGAGCAAACGAGG UA, GGACAGAAUAGCACGUAAA, CUAGAAGAGACGAAACGAA, GGACAAAGCUACAUGAUAG.

## Generation of cell lines

### STAG1-EGFP, EGFP-STAG1, STAG2-EGFP, EGFP-STAG1-RFP-STAG2

Homology arms (0.6–1.5 kb per arm) surrounding the start or stop codons of STAG1 and STAG2 were amplified from genomic DNA of HeLa Kyoto cells using primers identified by primer-blast (*Ye et al., 2012*) and cloned into vector pJet1.2 (Thermo Scientific K1232). EGFP or FLAG-mRFP coding sequences were introduced before the stop or after the start codon to generate homology-directed recombination (HDR) donor plasmids. CRISPR guide RNAs introducing nicks on either strand when bound to SpCas9(D10A) (*Ran et al., 2013*) were identified using crispr.mit.edu and cloned into plasmid pX335 (Addgene 42335). The following genomic sequences were targeted: ACAATACTTACTGTAACACtgg and TATTTTTTAAGGAAAATTTtgg (STAG1 N-terminus); TGAA-GAAAATTTACAAATCtgg and TCTTCAGACTTCAGAACATagg (STAG1 C-terminus); ATTTACG TGGGTAAAATGGtgg and GAATATATTTCTGACATTGagg (STAG2 N-terminus); CACAGATTTAA TTGTGTACtgg and CAGTACACAATTAAATCTGtgg (STAG2 C-terminus). HeLa Kyoto cells were transfected with two guide RNA and one HDR donor plasmid (or four guide RNA and two HDR donor plasmids for the EGFP-STAG1 FLAG-mRFP-STAG2 double tagged cell line) using Lipofect-amine 2000 (Invitrogen 11668019). Cells were grown for 7–10 days before sorting single cells into 96 well plates. Homozygous targeting of genomic alleles was assessed by PCR and by immunoblotting after fractionation (*Ladurner et al., 2016*).

### Scc1-LAP, Smc1-LAP

For generation of Scc1-LAP and Smc1-LAP immortalized mouse embryonic fibroblasts (iMEFs), primary mouse embryonic fibroblasts (pMEFs) were isolated from E13.5 embryos as described previously (*Michalska, 2007*). Immortalized mouse embryonic fibroblasts (iMEFs) were then generated by the 3T3 protocol. The LAP tag was introduced as described (*Poser et al., 2008*). Briefly, Smc3-LAP or Scc1-LAP BAC constructs were introduced using Fugene HD transfection reagents. Cells were then selected based on geneticin (G418) resistance and thereafter FACS sorted based on GFP expression levels.

### STAG1-AID , STAG2-AID

The HeLa Kyoto N-terminally-tagged SA1/SA2 auxin-inducible degron (AID) cell lines were created by CRISPR/Cas9 mediated genome editing as described previously (*Wutz et al., 2017*). The cloning primers that were used for generating EGFP-AID-STAG1 were GTCTTCAGACTTCAGAACA T, and GGTTTCTCATCATTTTTCTA. Primers used for genotyping were forward primer: GCCAGC TGGGAATCTCTTCA, and reverse primer: GCCACAGTTTGCTGACTCCT. The EGFP-AID-STAG2 cell line cloning primers were GCACAGATTTAATTGTGTAC, and GCTCTCTCTCATTAGGTTCT. The primers used for genotyping were forward primer: AGAAAGAAGGCAAGCCACCA and reverse primer: GGCAGCAGGAAGTACCTAACT.

## FRAP

For FRAP of SMC3-LAP, cells expressing fluorescent cohesin subunits were grown on chambered coverglass (Nunc 155409) for 1–3 days while treated with siRNAs and thymidine as indicated. Cells were imaged at 37°C on a Zeiss LSM5 duo confocal microscope with 63x Plan-Apochromat objective and a 488 nm 100 mW diode laser for bleaching, or on a Zeiss LSM780 confocal microscope with 63x objective and bleaching with argon and diode lasers at 488 and 561 nm for dual color FRAP using $CO_2$-independent media, or on an LSM880 confocal microscope (Carl Zeiss) with a 40 × 1.4 NA oil DIC Plan- Apochromat objective (Zeiss) in cell culture medium without riboflavin and phenol red at 5% $CO_2$. Cells were either cell cycle synchronized as described above, or G1 and G2 phase were identified by nuclear and cytoplasmic distribution of DHB-mKate2 signals, respectively.

Cycloheximide (1 ug/ml) was added before imaging to inhibit protein synthesis and contribution of new GFP expression to signal recovery.

For spot FRAP, a circular region (r = 2 um) was bleached three times. Recovery of fluorescence was recorded over 10 min and 300 frames at 2 s intervals and normalized to 10 pre-bleach frames and background and cellular fluorescence measured with Fiji (*Schindelin et al., 2012*). Recovery curves were analyzed using Berkeley Madonna (www.berkeleymadonna.com). Curves were fitted by an exponential function with variables for free and transiently chromatin associated (*Ladurner et al., 2014*), dynamically and stably chromatin bound cohesin (*Gerlich et al., 2006*). Relative fractions and their residence times (reciprocal of the dissociation constant) were averaged and plotted using Prism software (GraphPad).

Inverse FRAP was used to specifically measure dynamics of cohesin bound to chromatin over several hours. To this end for SMC3-LAP (which is present in the nucleus and cytoplasm), an area covering the cell body was bleached except for a semicircle corresponding to approximately half of the nucleus, and cells were imaged intermittently using a motorized stage. Recovery was recorded over 2–4 hr at 3 min intervals and normalized as above. Curve fitting with single and bi-exponential function was used to deduce relative fractions and residence times of dynamic and stable cohesin on chromatin and plotted as above (see https://github.com/rladurner/STAG1/blob/master/curvefit.ipynb; *Ladurner, 2020*; copy archived at https://github.com/elifesciences-publications/STAG1). For STAG1-EGFP and EGFP-STAG2 (which show only nuclear GFP signal), iFRAP photobleaching was performed in half of nuclear regions with 2 iterations of 488 nm laser at max intensity after acquisition of two images. Fluorescence was measured in bleached- and unbleached regions followed by background subtraction with 1 min interval. iFRAP curves were normalized to the mean of the pre-bleach fluorescent intensity and to the first image after photobleaching. Curve fitting was performed with single exponential functions $f(t)=EXP(-kOff1*t)$ or double exponential functions $f(t)=a*EXP(-kOff1*t)+(1-a)*EXP(-kOff2*t)$ in R using the minpack.lm package (version 1.2.1). Dynamic and stable residence times were calculated from $1/kOff1$ and $1/kOff2$ respectively. Double exponential curve fitting was performed under constraint that $1/kOff1$ and $1/kOff2$ are in range between 1 min-40 min and 1.5 hr-15 hr respectively. Soluble fractions were estimated by the reduction of fluorescence signals in unbleached area after photobleaching.

## Chromatin fractionation

Cells were extracted in a buffer consisting of 20 mM Tris-HCl (pH 7.5), 100 mM NaCl, 5 mM MgCl, 2 mM NaF, 10% glycerol, 0.2% NP40, 20 mM β-glycerophosphate, 0.5 mM DTT and protease inhibitor cocktail (Complete EDTA-free, Roche). Chromatin pellets and supernatant were separated and collected by centrifugation at 2,000 *g* for 5 min. The chromatin pellets were washed three times with the same buffer.

## Immunoprecipitation

Cells synchronized in S phase by a thymidine treatment were released for 16 hr to G1 phase and harvested. Pellets were resuspended in lysis buffer and used for single-step IP of chromatin-bound fractions. The beads coupled to antibody and bound to responding proteins were mildly washed, and captured proteins were eluted with 0.1 M glycine, and eluates were neutralized with 1.5 M Tris–HCl pH9.2.

## ChIP-seq

SMC3, GFP and CTCF ChIP-seq was performed as described in *Wendt et al. (2008)*. SMC3(ac) ChIP was performed as in *Schmidt et al. (2009)*. In brief, cell pellets were hypotonically treated, nuclei were isolated, lysed, and sonicated. Lysates were incubated overnight with protein-G dyna beads pre-bound to SMC3(ac) antibody (*Nishiyama et al., 2010*) or mouse non-immune IgG (SACSC-2025). Beads were washed, samples were eluted and de-crosslinked, DNA was purified and sequenced.

## Mass spectrometry

To generate a peptide spanning lysin 105 and 106 the immunoprecipitates were digested in solution with 400 ng Glu-C (Sequencing Grade, Roche) at 37°C for 16 hr.

The mass traces for the peptide VSLRRVIGAKKD in its unmodified, singly acetylated and doubly acetylated form were extracted from the raw files using the program Qualbrowser which is part of the Xcalibur software (Thermo Scientific). To account for different amounts of SMC3 protein between the immunoprecipitates the peptide area values were normalized based on the sum of the three most intense unmodified SMC3 peptides.

## DNA FISH

DNA probes (BACs, Fosmids and Cosmids) were ordered at BAC Resources PAC and purified by midi-prep purification kit (Quiagen). Probes were labelled by nick translation using 1–2 µg DNA per 50 µL reaction (Sigma Aldrich). Probes were fluorescently labelled using Alexa dyes (Alexa-488, Alexa-568). Every 5 µL of nick-translated probe was ethanol precipitated together with 1 µL of salmon sperm DNA, 3 µL human Cot-1, 0.5 µL 3M sodium acetate and 60 µL ethanol 100%. Probes were then snap frozen in liquid nitrogen and centrifuged at 13000 rpm for 10 min at four degrees. The supernatant was carefully removed and replaced with 200 µL of ethanol 70%, and the probe mixture was centrifuged at 13000 rpm for 10 min at four degrees. The supernatant was carefully removed and the pellet air dried protected from light. The pellet was resuspended in 5 µL hybridization buffer (2xSSC, 20% w/v dextran sulfate, 50% formamide pH7) at 37 degrees for 10 min then denatured for 7 min at 80 degrees and incubated at 37 degrees for 30 min before use. HeLa cells were cultured, siRNA treated and synchronized on coverslips. Cells were quickly rinsed with PBS three times and fixed in a solution of PBS with 4% paraformaldehyde for 10 min at room temperature. Cells were then washed twice in PBS for 5 min each. Permeabilization of cells was performed in freshly made PBS, 0.5 Triton X-100 for 7 min on ice. Cells were washed twice with a solution of ethanol 70% for 5 min and dehydrated in 80%, 95% and 100% ethanol for 3 min each. Cells were then air dried and denatured in 50% formamide, 2xSSC adjusted at pH 7.2 for 30 min at 80 degrees. Cells were washed three times in cold 2xSSC. The coverslip was placed cell-side down onto the prepared fluorescently labelled probes on a slide and sealed with glue. Hybridization was performed overnight at 42 degrees in a dark and humid chamber. The glue was then removed carefully, and coverslips were placed cell-side up and washed protected from light three times in warm 50% formamide, 2xSSC pH 7.2 for 5 min each at 42 degrees and three times in 2xSSC for 5 min. Cells were briefly washed in 2xSSC at room temperature and counterstain in 0.2 mg/mL DAPI solution for 2 min at room temperature and washed twice in 2xSSC for 5 min each. Coverslips were dried and mounted in Vectashield and fixed with a minimal amount of nail varnish. Acquisitions were performed on LSM 880 and 780 confocal microscopes. After tile scan, hundreds of single nuclei positions were spotted using DAPI channel in X, Y and Z. Three-dimensional acquisitions were then made for each of these positions for all channels and saved for processing.

Batch Alleles Investigation Tool 'BAIT' was designed to run on Definiens software and measure FISH three-dimensional inter-probe distances automatically. Nucleus segmentation was performed in three-dimensions by use of the auto-threshold function of Definiens on DAPI staining. Nuclear boundaries were analyzed to detect and discard incomplete nuclei in X, Y and Z. Probes signals were determined as pixel intensity values within the nucleus. To comparison size and intensity of the signals, geometrical measurements were performed to set an object center 'Seed' and clusters of spots 'Allele' for and between each channel. Spot center distances were then calculated in three dimensions and exported as comma-separated values (.csv). Quality control was performed under Definiens to detect aberrations and artefacts. Statistics were performed using R. Violin distribution plots were made with Prism 8. Figures were made with Fiji standard deviation projection.

## Comparison of FISH results with Hi-C maps

Numbers are given for the most densely populated square at highest resolution within the range of the loop coordinates targeted by our FISH probes. High-resolution Hi-C maps for control, STAG1-RNAi, and STAG2-RNAi at 5 Kb nominal resolution were generated using Juicebox Values indicate unique contact counts with KR (balanced) normalization, and no normalization applied.

## Length distribution, CTCF occupancy, and CTCF orientation of loops

The length of each loop was calculated as the distance between the midpoints of the loop anchors, and the length distributions of loops in each group were visualized with the empirical cumulative distribution function.

To determine CTCF occupancy, we expanded loop anchors smaller than 15 kb to 15 kb and counted the number of anchors that overlap with at least one CTCF ChIP-seq peak using bedtools. We calculated fold enrichment of CTCF occupancy by comparing the counts with the average overlap of ten random translational controls with the same length distribution as the loop anchors. Orientation of CTCF motifs at loop anchors was identified using MotifFinder in Juicer, and the proportion of inward oriented (following the convergent rule) CTCF motifs was reported.

## Aggregate peak analysis (APA) of loops

We performed aggregate peak analysis of *hiccups*-called loops using juicer_tools apa. The aggregate enrichment of our sets of looping peaks in contact matrices was visualized by plotting a cumulative stack of sub-matrices around detected loop coordinates. For a map with 10 Kb resolution we generated squares of 210 Kb x 210 Kb summing all putative loop peaks in a way that the resulting APA plot displays the total number of contacts which lie within the entire loop set at the center of the matrix within the aggregate pile-ups of their surroundings (*Rao et al., 2014*). All underlying matrices are KR-normalized.

## Aggregate analysis of TADs

For the analysis and visualization of average TAD pileups we generated size-sorted and -classified lists of TADs and calculated histogram matrices around the centers of those TAD areas within narrow predefined size ranges. That way several sub-matrices of interaction around similarly size-classified TADs were added up in order to generate a global profile of one size-range for every Hi-C matrix (10 Kb resolution, coverage-normalization).

## Hi-C insulation plots

Plots of the insulation scores were made based on insulation bedGraph files and TAD boundary coordinates generated by using the 'findTADsAndLoops.pl' script in the HOMER software package. This software scans relative contact matrices for locally dense regions of contacts or areas with an increased degree of intra-domain interactions relative to surrounding regions. Using a resolution of 3000, a window size of 15000, and the default maximum interaction distance (2 MB), we generated a coordinate set of sites with maximal transition in contact orientation, that is sites with highest insulation. Average plots of insulation profiles in all samples were made from regions cantered around the coordinates in the respective bedGraph files.

## Simulations

Molecular dynamic simulations were performed using HOOMD-blue (*Anderson et al., 2008*; *Glaser et al., 2015*) using an approach similar to *Sanborn et al. (2015)* with minor modifications. The region 17.65–19.25 mb on chromosome nine under wild type, STAG1-RNAi, and STAG2-RNAi conditions (*Figure 6B*) was simulated as a polymer of length 2000, each monomer representing 1 kb of chromatin, for a total of 850000 times steps. A total of 480 replicated simulations were performed and aggregated into a single contact map. All simulations contained an average of three cohesin complexes actively extruding chromatin to form loops. The probability for one end of the extrusion complex to halt at a particular locus was derived from CTCF ChIP-seq data in HeLa cells (ENCODE phase 2, Broad Institute, file ENCFF000BAN), normalized to a probability between 0 and 1. Each halted end of an extrusion complex also had a 0.05% probability to continue sliding at each time step (halting lifetime of ~2000 time steps). In addition, extrusion complexes have had predefined average lifetime on chromatin, and changes in STAG1 and STAG2 levels were simulated by modulating the extrusion lifetime. Wild type extrusion complexes had an average lifetime of 5000 timesteps (0.02% chance of dissociating at each time step) while STAG1-RNAi had 5-fold higher lifetimes and STAG2-RNAi had 2-fold lower lifetimes.

Simplified simulations of short, medium, or long extrusion lifetimes (*Figure 6A*) were performed similarly. Chromatin was represented as polymers of length 1000 containing an average of three

extrusion complexes. The two outermost CTCF binding sites had halting probabilities of 0.9 while the inner four binding sites had halting probabilities of 0.4. Halted extrusion ends had a 0.01% probability to continue sliding. Short, medium, and long extrusion lifetimes were modeled as complexes with lifetimes of 312.5, 625, and 10000-time steps respectively.

### ChIP-seq peak calling and calculation of peak overlaps

Peaks were called by the MACS algorithm version 1.4.2 (*Zhang et al., 2008*), using a P-value threshold of 1e-10 and by using sample and input read files. We identified sites of overlapping peaks between different conditions as well as between SMC3 and CTCF peaks using the MULTOVL software (*Aszódi, 2012*). We applied an inclusive type of overlap display ('union'), in which coordinates of overlapping peaks are merged into one common genomic site.

### Hi-C library preparation

We generated a total of 14 in situ Hi-C libraries from our RNAi experiments (*Supplementary file 1*). Libraries generated by using MboI enzyme were done as described in *Rao et al. (2014)* without modification. In brief, the in situ Hi-C protocol involves crosslinking cells with formaldehyde, permeabilizing nuclei with detergent, digesting DNA overnight using a 4-cutter restriction enzyme, filling in 5'-overhangs while incorporating a biotinylated nucleotide, ligating newly blunted ends together, shearing DNA, capturing biotinylated ligation junctions with streptavidin beads, and analysing the resulting fragments with paired-end sequencing. All the libraries generated with the 6 bp cutter HindIII were performed as in *Wutz et al. (2017)*.

### Hi-C data processing

All Hi-C libraries were sequenced with 150 bp paired-end reads and the resulting data was processed using Juicer (*Durand et al., 2016b*; *Rao et al., 2014*). These data were aligned against the hg19 reference genome. All contact matrices used for analysis were Knight-Ruiz or Vanilla-Coverage normalized with Juicer.

Loops were annotated in our RNAi experiments using *hiccups*(*Durand et al., 2016a*; *Rao et al., 2014*). Default parameters as described in *Durand et al. (2016a)*; *Rao et al. (2014)* were used to call loops at 5 kb and 10 kb resolutions and merged as described in *Rao et al. (2014)*. Domains were annotated in our RNAi experiments using Arrowhead (*Durand et al., 2016a*; *Rao et al., 2014*). Domains were called at 5 kb and 10 kb resolutions using default parameters and merged.

## Acknowledgements

Research in the laboratory of J-MP was supported by Boehringer Ingelheim, the Austrian Science Fund (FWF special research program SFB F34 'Chromosome Dynamics' and Wittgenstein award Z196-B20), the Austrian Research Promotion Agency (Headquarter grant FFG-852936, the European Community's Seventh Frame- work Programme (FP7/2007-2013) under grant agreement 241548 (MitoSys), the European Research Council (ERC) under the European Union's Horizon 2020 research and innovation programme GA No 693949, and by Human Frontier Science Program RGP0057/2018. The work in the lab of KM has been supported by EPIC-XS, project number 823839, funded by the Horizon 2020 program of the European Union, and the Austrian Science Fund by ERA-CAPS I 3686 International Project. Research in the laboratory of ELA was supported by NIH (5UM1HG009375-03 and 5U01HL130010-05) and NSF (PHY-1427654). Research in the laboratory of PF was supported by ERC Advanced Grant 340152 DEVOCHROMO and Biotechnology and Biological Sciences Research Council (BB/J004480/1).

## Additional information

### Funding

| Funder | Grant reference number | Author |
| --- | --- | --- |
| NIH Clinical Center | 5U01HL130010-05 | Erez Lieberman-Aiden |
| NIH Clinical Center | 5UM1HG009375-03 | Erez Lieberman-Aiden |

| National Science Foundation | PHY-1427654 | Erez Lieberman-Aiden |
|---|---|---|
| Horizon 2020 Framework Programme | EPIC-XS 823839 | Karl Mechtler |
| Austrian Science Fund | 3686 International Project | Karl Mechtler |
| H2020 European Research Council | 693949 | Jan-Michael Peters |
| Vienna Science and Technology Fund | WWTF LS09-13 | Jan-Michael Peters |
| Austrian Science Fund | SFB F34 | Jan-Michael Peters |
| Österreichische Forschungsförderungsgesellschaft | FFG-852936 | Jan-Michael Peters |
| Austrian Science Fund | Z196-B20 Wittgenstein award | Jan-Michael Peters |
| Seventh Framework Programme | FP7/2007-2013 241548 | Jan-Michael Peters |
| Human Frontier Science Program | RGP0057/2018 | Jan-Michael Peters |
| European Research Council | 340152 DEVOCHROMO | Peter Fraser |
| Biotechnology and Biological Sciences Research Council | BB/J004480/1 | Peter Fraser |
| EMBO | Long Term Fellowship ALTF 1335-2016 | Kota Nagasaka |
| Human Frontier Science Program | Fellowship LT001527/2017 | Kota Nagasaka |
| Boehringer Ingelheim | | Jan-Michael Peters |

The funders had no role in study design, data collection and interpretation, or the decision to submit the work for publication.

## Author contributions

Gordana Wutz, Conceptualization, Data curation, Formal analysis, Validation, Investigation, Visualization, Writing - original draft, Writing - review and editing; Rene Ladurner, Conceptualization, Validation, Investigation, Visualization, Methodology; Brian Glenn St Hilaire, Miroslav P Ivanov, Stefan Schoenfelder, Petra van der Lelij, Validation, Investigation; Roman R Stocsits, Data curation, Formal analysis, Visualization; Kota Nagasaka, Formal analysis, Validation, Investigation, Visualization; Benoit Pignard, Conceptualization, Formal analysis, Validation, Investigation; Adrian Sanborn, Software, Formal analysis, Visualization, Methodology; Wen Tang, Resources, Validation, Investigation; Csilla Várnai, Xingfan Huang, Gerhard Dürnberger, Formal analysis, Visualization; Elisabeth Roitinger, Investigation, Visualization; Karl Mechtler, Peter Fraser, Supervision, Funding acquisition, Project administration; Iain Finley Davidson, Writing - original draft, Writing - review and editing; Erez Lieberman-Aiden, Conceptualization, Supervision, Funding acquisition, Validation; Jan-Michael Peters, Conceptualization, Supervision, Funding acquisition, Methodology, Writing - original draft, Project administration, Writing - review and editing

## Author ORCIDs

Gordana Wutz (ID) https://orcid.org/0000-0002-6842-0795
Kota Nagasaka (ID) http://orcid.org/0000-0003-0765-638X
Miroslav P Ivanov (ID) https://orcid.org/0000-0001-9352-0969
Stefan Schoenfelder (ID) https://orcid.org/0000-0002-3200-8133
Jan-Michael Peters (ID) https://orcid.org/0000-0003-2820-3195

## Decision letter and Author response

Decision letter https://doi.org/10.7554/eLife.52091.sa1
Author response https://doi.org/10.7554/eLife.52091.sa2

# Additional files

## Supplementary files

• Supplementary file 1. Summary statistics for Hi-C data sets generated in this study. A. Number of the library. B. Condition used to generate the library. C. Number of the biological replicate. D. Restriction enzyme used to generate the Hi-C library. E. Raw number of read pairs from paired-end sequencing. F. Unique valid mapped read pairs from HiCUP v0.7.1. G. Number of unique valid read pairs that are inter-chromosomal. H. Percentage of unique valid read pairs that are inter-chromosomal. I. Log2 contact enrichment of A-A and B-B contacts for long-range (>10 Mb) intra-chromosomal contacts. J. Log2 contact enrichment of A-A and B-B contacts for inter-chromosomal contacts, K. Percentage of genome covered by TADs called by HOMER v4.7. L. Number of TADs called by HOMER v4.7. M. Number of loops called by the *hiccups* algorithm of Juicer tools v0.7.5. N. Average standardized insulation score at the corresponding G1 control TAD boundaries (hires or r1, r2 average) called by HOMER v4.7 in the respective conditions. O. Average standardized insulation score at the TAD boundaries called by HOMER v4.7 in the respective conditions. P. Number of loops called by the *hiccups* algorithm of Juicer tools v0.7.5; please note that the number of loops that can be called depends on the number of unique read pairs. This needs to be taken into consideration when comparing corner peaks between different experiments.

• Supplementary file 2. Number of cells analyzed by FISH and statistical significance. Number of cells analyzed by FISH in *Figure 5—figure supplement 2* for control,CTCF, SCC1, STAG1, STAG2 and double STAG1/STAG2 RNAi. Statistical significance is measured by t-test relative to the control.

• Transparent reporting form

## Data availability

Sequencing data have been deposited in GEO under accession code GSE138405, and is available at https://www.ncbi.nlm.nih.gov/geo/query/acc.cgi?acc=GSE138405.

The following dataset was generated:

| Author(s) | Year | Dataset title | Dataset URL | Database and Identifier |
|---|---|---|---|---|
| Wutz G, Ladurner R, St Hilaire B, Stocsits R, Nagasaka K, Pignard B, Sanborn A, Tang W, Várnai C, Ivanov M, Schoenfelder S, van der Lelij P, Huang X, Dürnberger G, Roitinger E, Mechtler K, Davidson IF, Fraser P, Aiden EL, Peters JM | 2020 | ESCO1 and CTCF enable formation of long chromatin loops by protecting cohesinSTAG1 from WAPL | https://www.ncbi.nlm.nih.gov/geo/query/acc.cgi?acc=GSE138405 | NCBI Gene Expression Omnibus, GSE138405 |

The following previously published dataset was used:

| Author(s) | Year | Dataset title | Dataset URL | Database and Identifier |
|---|---|---|---|---|
| Gordana Wutz, Roman R Stocsits | 2017 | Topologically associating domains and chromatin loops depend on cohesin and are regulated by CTCF, WAPL and PDS5 proteins | https://www.ncbi.nlm.nih.gov/geo/query/acc.cgi?acc=GSE102884 | NCBI Gene Expression Omnibus, GSE102884 |

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
