## [Decision Letter]

**Decision letter after peer review:**

Thank you for submitting your article "ESCO1 and CTCF enable formation of long chromatin loops by protecting cohesin^STAG1^ from WAPL" for consideration by *eLife*. Your article has been reviewed by a Senior Editor, a Reviewing Editor, and two reviewers. The following individuals involved in review of your submission have agreed to reveal their identity: Hongtao Yu (Reviewer #2).

We find your manuscript of interest and would like to invite you to prepare a revised submission. The reviewers have provided thoughtful comments and we would like to see some of their concerns addressed directly in a revised manuscript.

One main concern is that you may have overstated the conclusion that ESCO1 and CTCF promote the loop-forming ability of cohesin-STAG1 through antagonizing WAPL. Reviewer 2 suggests that this is more of an inference from what is currently known about WAPL. You may wish to address this concern by adding more epistasis analyses among ESCO1, CTCF, and WAPL. Or else you might consider toning down the conclusion.

We also agree that the dual use of TADs and loops will be confusing for the reader. They are not synonymous and should be treated differently. Please define their usage clearly and reference the terms as appropriate in the text.

Finally, we also agree that critical information in the supplement should be moved to the main figures where possible. *eLife* has a flexible format which allows you to present more data in the main figures.

Reviewer #1:

The manuscript describes an analysis of the separate roles of STAG1, STAG2, CTCF, SCO1, and WAPL in the formation of strong point-to-point interactions manifested as loops in Hi-C experiments. Results suggest that the residence time of these various proteins and their acetylated forms explain their ability to form loops of different lengths and strengths by cohesin extrusion.

The results presented in the manuscript are significant because they offer mechanistic insights into how different forms of cohesin can form different types of interactions. Although not addressed in the manuscript, this may translate into different functional outcomes for various loop types in the regulation of enhancer-promoter interactions. The results are significant and of broad interest, and the manuscript is suitable for publication in *eLife*.

1) Too much important information is presented in the supplemental figures. I'm not sure what the restrictions for numbers of figures are in *eLife* but, if possible, authors should move some of the information from supplemental to main figures.

2) Simultaneous use of the terms "TADs" and "loops" is confusing. Are the two domains the same or different? Since authors are only talking about CTCF/cohesin-mediated structures, perhaps they should only use the term loop. If "TAD" is used to refer to a different type of domain, authors should explain how this domain is different from CTCF/cohesin loops.

3) In subsection “Long-range chromatin interactions mediated by cohesinSTAG1 are similar but not identical to those observed in WAPL depleted cells”, "loop calling by "Juicer tool". Do you mean HiCCUPS?

Reviewer #2:

The ring-shaped cohesin complex organizes the genome into loops and topology associated domains (TADs), thereby regulating transcription and recombination. Cohesin also entraps sister chromatids to mediate genome segregation during cell division. One of the four core cohesin subunits has two paralogs, STAG1 and STAG2, in vertebrate cells. STAG1 and STAG2 have somewhat redundant functions for sister-chromatid cohesion during cell division, but have non-redundant functions during development. Whether and how the two forms of cohesin perform different functions during interphase are unclear. In the current study, Wutz et al. show that cohesin-STAG1 is preferentially acetylated on its SMC3 subunit, has longer resident times on chromatin, and mediates the formation of longer chromosome loops. By contrast, cohesin-STAG2 is more dynamic on chromatin and mediates the formation of smaller loops. They further show that the long loops formed by cohesin-STAG1 requires the acetyltransferase ESCO1 and the chromatin insulator CTCF, but not the cohesin-stabilization factor SORORIN. These data suggest that cohesin might organize the interphase genome through the formation of nested loops.

In addition to this main conclusion, the authors present evidence to suggest several other interesting points: (1) WAPL might directly contribute to the enrichment of cohesin at convergent CTCF sites; (2) CTCF promotes cohesin-STAG1 acetylation in G1; (3) ESCO1 counteracts compartmentalization of chromatin. These points, though not proven, can spur further investigations in the field.

Overall, I feel that the paper constitutes an important contribution to cohesin biology and should be published in *eLife*. The following major comments need to be addressed prior to publication.

Major comments:

1) The title suggests that ESCO1 and CTCF promote the loop-forming ability of cohesin-STAG1 through antagonizing WAPL. There are, however, not that many pieces of data in the manuscript that directly support this conclusion. It is more of an inference from current knowledge about WAPL regulation. The authors may wish to add more epistasis analyses among ESCO1, CTCF, and WAPL to bolster their claim. Alternatively, they can tone down their conclusion.

2) The HiC maps of STAG2 RNAi and STAG2/ESCO1 RNAi cells in Figure 6A are not that visibly different. Yet, the loop calling algorithms detect major changes. Can the authors comment on why this is? What is the key feature that the algorithms detect in the loops but is not apparent to the human eye?

---

## [Author Response]

Reviewer #1:The manuscript describes an analysis of the separate roles of STAG1, STAG2, CTCF, SCO1, and WAPL in the formation of strong point-to-point interactions manifested as loops in Hi-C experiments. Results suggest that the residence time of these various proteins and their acetylated forms explain their ability to form loops of different lengths and strengths by cohesin extrusion.The results presented in the manuscript are significant because they offer mechanistic insights into how different forms of cohesin can form different types of interactions. Although not addressed in the manuscript, this may translate into different functional outcomes for various loop types in the regulation of enhancer-promoter interactions. The results are significant and of broad interest, and the manuscript is suitable for publication in eLife.1) Too much important information is presented in the supplemental figures. I'm not sure what the restrictions for numbers of figures are in eLife but, if possible, authors should move some of the information from supplemental to main figures.

We have followed the reviewer’s recommendation by rearranging the figures as follows:

Figure S8A, B and C has become Figure 4C, D and F.

Figure S9D has become Figure 5A.

Parts of Figure S12C and E have become Figure7B and C.

2) Simultaneous use of the terms "TADs" and "loops" is confusing. Are the two domains the same or different? Since authors are only talking about CTCF/cohesin-mediated structures, perhaps they should only use the term loop. If "TAD" is used to refer to a different type of domain, authors should explain how this domain is different from CTCF/cohesin loops.

We agree with the reviewer that the current terminology in the literature is confusing as the term “loops” is used both generically to describe long-range chromosomal *cis* interactions, including those that TADs are composed of, as well as specific “loop” features in Hi-C heat maps that are also referred to as “dots” and “corner peaks”. We assume that at the molecular level TADs are composed of the same long-range interactions as loops and therefore agree that the term loop should be used as widely as possible. However, it is not possible to avoid the term TAD entirely as it represents a characteristic feature of Hi-C maps that needs to be analyzed and described.

To address these issues, we now amended the text by (i) explaining the terminology more explicitly in the first paragraph of the Introduction and (ii) by carefully using appropriate and specific wording throughout the text, i.e. we are only using the term TAD when describing characteristic pyramid-shaped structures in Hi-C maps.

3) In subsection “Long-range chromatin interactions mediated by cohesinSTAG1 are similar but not identical to those observed in WAPL depleted cells”, "loop calling by "Juicer tool". Do you mean HiCCUPS?

We thank the reviewer for pointing out this imprecise description. Yes, we meant the HiCCUPs algorithm which is part of the Juicer Tool software suite and have changed the text accordingly.

Reviewer #2:The ring-shaped cohesin complex organizes the genome into loops and topology associated domains (TADs), thereby regulating transcription and recombination. Cohesin also entraps sister chromatids to mediate genome segregation during cell division. One of the four core cohesin subunits has two paralogs, STAG1 and STAG2, in vertebrate cells. STAG1 and STAG2 have somewhat redundant functions for sister-chromatid cohesion during cell division, but have non-redundant functions during development. Whether and how the two forms of cohesin perform different functions during interphase are unclear. In the current study, Wutz et al. show that cohesin-STAG1 is preferentially acetylated on its SMC3 subunit, has longer resident times on chromatin, and mediates the formation of longer chromosome loops. By contrast, cohesin-STAG2 is more dynamic on chromatin and mediates the formation of smaller loops. They further show that the long loops formed by cohesin-STAG1 requires the acetyltransferase ESCO1 and the chromatin insulator CTCF, but not the cohesin-stabilization factor SORORIN. These data suggest that cohesin might organize the interphase genome through the formation of nested loops.In addition to this main conclusion, the authors present evidence to suggest several other interesting points: (1) WAPL might directly contribute to the enrichment of cohesin at convergent CTCF sites; (2) CTCF promotes cohesin-STAG1 acetylation in G1; (3) ESCO1 counteracts compartmentalization of chromatin. These points, though not proven, can spur further investigations in the field.Overall, I feel that the paper constitutes an important contribution to cohesin biology and should be published in eLife. The following major comments need to be addressed prior to publication.Major comments:1) The title suggests that ESCO1 and CTCF promote the loop-forming ability of cohesin-STAG1 through antagonizing WAPL. There are, however, not that many pieces of data in the manuscript that directly support this conclusion. It is more of an inference from current knowledge about WAPL regulation. The authors may wish to add more epistasis analyses among ESCO1, CTCF, and WAPL to bolster their claim. Alternatively, they can tone down their conclusion.

The reviewer is correct. We have therefore now performed epistasis experiments in which we analyzed the effects of CTCF and WAPL depletion on the chromatin residence time of cohesin^STAG1^. These experiments revealed that the decrease in chromatin residence time that is caused by CTCF depletion is reverted by co-depletion of WAPL, as is shown in the new Figure 3F and G. These results support our hypothesis that CTCF protects cohesin^STAG1^ from release by WAPL.

For technical reasons we have not yet been able to perform the equivalent epistasis experiments for ESCO1 and WAPL. Since these experiments would require significantly more time, we have decided to town down our conclusions in this case.

2) The HiC maps of STAG2 RNAi and STAG2/ESCO1 RNAi cells in Figure 6A are not that visibly different. Yet, the loop calling algorithms detect major changes. Can the authors comment on why this is? What is the key feature that the algorithms detect in the loops but is not apparent to the human eye?

We agree that the Hi-C maps previously shown in Figure 6A (Figure 7—figure supplement 1C in the revised version) do not appear as different as their analysis by the HiCCUPs loop calling algorithm suggests. We suspect that this impression is caused by the resolution of the compressed images in which the Hi-C maps are shown in the figure, which is much lower than the resolution of the Hi-C maps that are analyzed by HiCCUPs at the level of individual pixels which are being compared for intensity differences.